# Closing the Gap: Tighter Analysis of Alternating Stochastic Gradient Methods for Bilevel Problems

**Tianyi Chen**
Rensselaer Polytechnic Institute
chentianyi19@gmail.com

**Yuejiao Sun**
UCLA
sunyj@math.ucla.edu

**Wotao Yin**
UCLA
wotaoyin@math.ucla.edu

## Abstract

Stochastic nested optimization, including stochastic bilevel, min-max, and compositional optimization, is gaining popularity in many machine learning applications. While the three problems share a nested structure, existing works often treat them separately, thus developing problem-specific algorithms and analyses. Among various exciting developments, simple SGD-type updates (potentially on multiple variables) are still prevalent in solving this class of nested problems, but they are believed to have a slower convergence rate than non-nested problems. This paper unifies several SGD-type updates for stochastic nested problems into a single SGD approach that we term ALternating Stochastic gradient dEscenT (ALSET) method. By leveraging the *hidden smoothness* of the problem, this paper presents a tighter analysis of ALSET for stochastic nested problems. Under the new analysis, to achieve an $\epsilon$-stationary point of the nested problem, it requires $\mathcal{O}(\epsilon^{-2})$ samples in total. Under certain regularity conditions, applying our results to stochastic compositional, min-max, and reinforcement learning problems either improves or matches the best-known sample complexity in the respective cases. *Our results explain why simple SGD-type algorithms in stochastic nested problems all work very well in practice without the need for further modifications.*

## 1 Introduction

Stochastic gradient descent (SGD) methods [1] are prevalent in solving large-scale machine learning problems. Often, SGD is applied to solve stochastic problems with a relatively simple structure. Specifically, applying SGD to minimize the function $\mathbb{E}_\xi\left[f(x;\xi)\right]$ over the variable $x \in \mathbb{R}^d$, we have the iterative update $x^{k+1} = x^k - \alpha\nabla f(x^k;\xi^k)$, where $\alpha > 0$ is the stepsize and $\nabla f(x^k;\xi^k)$ is the stochastic gradient at the iterate $x^k$ and the sample $\xi^k$. However, many problems in machine learning today, such as meta learning, deep learning, hyper-parameter optimization, and reinforcement learning, go beyond the above simple minimization structure (termed the non-nested problem thereafter). For example, the objective function may be the compositions of multiple functions, where each composition may introduce an additional expectation [2]; and, the objective function may depend on the solution of another optimization problem [3]. In these problems, how to apply SGD and the efficiency of running SGD are not fully understood.

To answer these questions, in this paper, we consider the following form of *stochastic nested optimization problems*, which is a generalization of the non-nested problems, given by

$$\min_{x\in\mathbb{R}^d} \quad F(x) := \mathbb{E}_\xi\left[f\left(x, y^*(x); \xi\right)\right] \qquad \text{(upper)} \qquad (1a)$$

$$\text{s.t.} \quad y^*(x) = \arg\min_{y\in\mathbb{R}^{d'}} \mathbb{E}_\phi[g(x, y; \phi)] \qquad \text{(lower)} \qquad (1b)$$

where $f$ and $g$ are differentiable functions; and, $\xi$ and $\phi$ are random variables. In the optimization literature [4–6], the problem (1) is referred to as the stochastic *bilevel* problem, where the upper-level

35th Conference on Neural Information Processing Systems (NeurIPS 2021).

optimization problem depends on the solution of the lower-level optimization over $y \in \mathbb{R}^{d'}$, denoted as $y^*(x)$, which depends on the value of upper-level variable $x \in \mathbb{R}^d$.

The stochastic *bilevel* nested problem (1) encompasses two popular formulations with the nested structure: stochastic min-max problems and stochastic compositional problems. Therefore, results on the general nested problem (1) will also imply the results in the special cases. For example, if the lower-level objective $g$ is the negative of the upper-level objective $f$, i.e., $g(x, y; \phi) := -f(x, y; \xi)$, the stochastic bilevel problem (1) reduces to the stochastic min-max problem

$$\text{If } g(x, y; \phi) := -f(x, y; \xi) \quad \Rightarrow \quad \min_{x \in \mathbb{R}^d} F(x) := \max_{y \in \mathbb{R}^{d'}} \mathbb{E}_\xi \left[ f(x, y; \xi) \right]. \tag{2}$$

Motivated by applications in zero-sum games, adversarial learning and training GANs, significant efforts have been recently made for solving the stochastic min-max problem; see e.g., [7–11].

For example, if the upper-level objective $f$ is only a function of $y$, i.e., $f(x, y; \xi) := f(y; \xi)$, and the lower-level objective $g$ is a quadratic function of $y$, i.e., $g(x, y; \phi) := \|y - h(x; \phi)\|^2$ with a smooth function $h$ of $x$, then the variable $y^*(x)$ admits a closed-form solution, and thus the stochastic bilevel problem (1) reduces to the stochastic compositional problem [12–14]

$$\text{If } g(x, y; \phi) := \|y - h(x; \phi)\|^2 \quad \Rightarrow \quad \min_{x \in \mathbb{R}^d} F(x) := \mathbb{E}_\xi \left[ f\left( \mathbb{E}_\phi[h(x; \phi)]; \xi \right) \right]. \tag{3}$$

Stochastic compositional problems in the form of (3) have been studied in the applications in model-agnostic meta learning and policy evaluation in reinforcement learning; see e.g., [2, 15].

To solve the nested problem (1) by SGD, one natural solution is to apply *alternating SGD updates* on $x$ and $y$ based on their stochastic gradients

$$y^{k+1} = y^k - \beta_k h_g^k \quad \text{and} \quad x^{k+1} = x^k - \alpha_k h_f^k \tag{4}$$

where $h_g^k$ is the unbiased stochastic gradient of $\mathbb{E}_\phi[g(x^k, y^k; \phi)]$ and $h_f^k$ is the (possibly biased) stochastic gradient of $F(x^k)$; and, $\beta_k$ and $\alpha_k$ are the stepsizes. A key challenge of running (4) for the nested problem is that (stochastic) gradient of the upper-level variable $x$ is prohibitively expensive to compute. As we will show later, computing an unbiased stochastic gradient of $F(x)$ requires solving the lower-level problem exactly to obtain $y^*(x)$.

An accurate stochastic gradient $h_f^k$ can be obtained in roughly three ways. One way is to run SGD updates on $y^k$ multiple times before updating $x^k$, which yields a double-loop algorithm. To guarantee convergence, it typically requires either the increasing number of lower-level $y$-update or the growing number of batch size to estimate $h_g^k$; see e.g., [16, 17]. The second way is to update $y^k$ in a timescale faster than that of $x^k$ so that $x^k$ is relatively static with respect to $y^k$; i.e., $\lim_{k \to \infty} \alpha_k / \beta_k = 0$; see e.g., [18]. The third way is to modify the direction $h_g^k$ of $y^k$ by incorporating additional correction term, which adds extra computation burden; see e.g., [19]. At a high level, these modifications either deviate from the lightweight implementation of SGD or sacrifice the sample complexity of SGD.

To this end, the **main goal** of this paper is to study the efficiency of running the vanilla alternating SGD (4) for the nested problem (1) and its implications on the special problem classes (2)-(3).

## 1.1 Main results

This paper analyzes a unifying algorithm for the stochastic bilevel problems that runs SGD on each variable alternatingly. We provide sample complexity that matches the complexity of SGD for single-level stochastic problems. Our results explain why SGD-type algorithms in stochastic bilevel, min-max, and compositional problems work very well in practice without modifications, including correction, increasing batch size, and two-timescale stepsizes.

In the context of existing methods, our contributions can be summarized as follows.

C1) We connect three different classes of stochastic nested optimization problems (stochastic compositional, min-max, and bilevel optimization), and unify three popular SGD-type updates for the respective problems into a single SGD-type method. We call it the ALternating Stochastic gradient dEscenT (ALSET) method.

| | ALSET | BSA | TTSA | stocBiO | STABLE | SUSTAIN/RSVRB |
|---|---|---|---|---|---|---|
| **batch size** | $\mathcal{O}(1)$ | $\mathcal{O}(1)$ | $\mathcal{O}(1)$ | $\mathcal{O}(\epsilon^{-1})$ | $\mathcal{O}(1)$ | $\mathcal{O}(1)$ |
| **$y$-update** | SGD | $\mathcal{O}(\epsilon^{-1})$ SGD steps | SGD | SGD | correction | momentum |
| **samples in $\xi$** | $\mathcal{O}(\kappa^5\epsilon^{-2})$ | $\mathcal{O}(\kappa^6\epsilon^{-2})$ | $\mathcal{O}(\kappa^p\epsilon^{-\frac{5}{2}})$ | $\mathcal{O}(\kappa^5\epsilon^{-2})$ | $\mathcal{O}(\kappa^p\epsilon^{-2})$ | $\mathcal{O}(\kappa^p\epsilon^{-\frac{3}{2}})$ |
| **samples in $\phi$** | $\widetilde{\mathcal{O}}(\kappa^9\epsilon^{-2})$ | $\widetilde{\mathcal{O}}(\kappa^9\epsilon^{-3})$ | $\widetilde{\mathcal{O}}(\kappa^p\epsilon^{-\frac{5}{2}})$ | $\widetilde{\mathcal{O}}(\kappa^9\epsilon^{-2})$ | $\mathcal{O}(\kappa^p\epsilon^{-2})$ | $\widetilde{\mathcal{O}}(\kappa^p\epsilon^{-\frac{3}{2}})$ |

Table 1: Sample complexity of stochastic bilevel algorithms (BSA in [16], TTSA in [18], stocBiO in [17], STABLE in [19], SUSTAIN in [25], RSVRB in [26]) to achieve an $\epsilon$-stationary point of $F(x)$; the notation $\widetilde{\mathcal{O}}(\cdot)$ hides the terms of $\log\epsilon^{-1}$; the notation $\kappa^p$ denotes a polynomial function of $\kappa$ since the dependence on $\kappa$ is not explicit in [18, 19, 25, 26].

C2) Under the same assumptions made in most of the previous work, we discover that the solution of the lower-level problem is smooth – a property that is overlooked by the previous analyses. By leveraging the *hidden smoothness*, we present a tighter analysis of ALSET for the stochastic bilevel problems. Under the new analysis, to achieve an $\epsilon$-stationary point of the nested problem, ALSET requires $\mathcal{O}(\epsilon^{-2})$ samples in total, rather than the $\mathcal{O}(\epsilon^{-5/2})$ sample complexity in the existing literature.

C3) We further customize the analysis to the two special cases – the compositional and min-max problems, and establish the improved sample complexity relative to that in the literature. We apply a new analysis to the celebrated actor-critic method for reinforcement learning problems. Under some regularity conditions, we show that, to achieve an $\epsilon$-stationary point, the single-loop actor-critic method requires $\mathcal{O}(\epsilon^{-2})$ samples with i.i.d. sampling, which improves the best-known result of $\mathcal{O}(\epsilon^{-5/2})$ in the literature.

## 1.2 Other related works

To put our work in context, we review prior art that we group in the following three categories.

**Stochastic bilevel optimization.** We can trace the study of bilevel optimization to the 1950s [20]. Many recent efforts have been made to solve the bilevel problems. One successful approach is to reformulate the bilevel problem as a single-level problem by replacing the lower-level problem by its optimality conditions [4, 5]. Recently, gradient-based methods for bilevel optimization have gained popularity. They iteratively approximate the (stochastic) gradient of the upper-level problem either in a forward or backward manner [21, 3, 22, 23]. Recent work has also studied the case where the lower-level problem does not have a unique solution [24].

The non-asymptotic analysis of bilevel optimization algorithms has been recently studied in some *pioneering* works, e.g., [16, 18, 17], just to name a few. In both [16, 17], bilevel stochastic optimization algorithms have been developed that run in a double-loop manner. To achieve an $\epsilon$-stationary point, they only need the sample complexities $\mathcal{O}(\epsilon^{-3})$ and $\mathcal{O}(\epsilon^{-2})$, respectively, comparable to that of SGD for the single-level case. Recently, a single-loop two-timescale stochastic approximation algorithm has been developed in [18] for the bilevel problem (1). Due to the nature of the two-timescale update, it incurs the sub-optimal sample complexity $\mathcal{O}(\epsilon^{-5/2})$. A single-loop single-timescale stochastic bilevel optimization method has been recently developed in [19]. While the method can achieve the sample complexity $\mathcal{O}(\epsilon^{-2})$, the resultant update on $y$ needs extra matrix projection, which can be costly. Very recently, the momentum-based acceleration has been incorporated into both the $x$- and $y$-updates in [25, 26] and also in [27] after our submission to the conference, where the new algorithms therein enjoy an improved sample complexity $\mathcal{O}(\epsilon^{-3/2})$. However, these results cannot imply the $\mathcal{O}(\epsilon^{-2})$ sample complexity of the alternating SGD update (4), and are orthogonal to our results. A comparison of our results with prior work can be found in Table 1.

**Stochastic min-max optimization.** In the context of min-max problems, the alternating version of the stochastic gradient descent ascent (GDA) method can be viewed as the alternating SGD updates (4) for the special nested problem (2). To mitigate the cycling behavior of GDA for convex-concave min-max problems, several variants have been developed by incorporating the idea of optimism; see e.g., [7, 8, 11, 29]. The analysis of stochastic GDA in the nonconvex-strongly concave setting is closely related to this paper; e.g., [9, 10, 30, 28]. Specifically, for stochastic GDA (SGDA), the $\mathcal{O}(\epsilon^{-2})$ sample complexity has been established in [28] under an increasing batch size $\mathcal{O}(\epsilon^{-1})$. As highlighted in [28], how to achieve the $\mathcal{O}(\epsilon^{-2})$ sample complexity under an $\mathcal{O}(1)$ constant batch size

remains open. The reduction of our results to the min-max setting will provide an answer to this open question. In the same setting, accelerated GDA algorithms have been developed in [31–33]. Going beyond the one-side concave settings, algorithms and their convergence analysis have been studied for nonconvex-nonconcave min-max problems with certain benign structure; see e.g., [8, 34–36]. A comparison of our results with prior work can be found in Table 2.

**Stochastic compositional optimization.** Stochastic compositional gradient algorithms developed in [12, 37] can be viewed as the alternating SGD updates (4) for the special compositional problem (3). However, to ensure convergence, the algorithms [12, 37] use two sequences of variables being updated in two different time scales, and thus the complexity of [12] and [37] is worse than $\mathcal{O}(\epsilon^{-2})$ of SGD for the non-compositional case. While most of existing algorithms rely on either two-timescale updates, the single-timescale single-loop approaches have been recently developed in [14, 38, 39], which achieve the sample complexity $\mathcal{O}(\epsilon^{-2})$, same as SGD for the non-nested problems. However, the algorithms proposed therein are not the vanilla alternating SGD update in the sense of (4). Other related compositional algorithms also include [40–42]. A comparison can be found in Table 3.

**Organization.** The basic background of bilevel optimization is reviewed, and the tighter analysis of the unifying ALSET method is presented in Section 2. The reduction of the main results to the special stochastic nested problems is provided in Section 3, and its applications to the actor-critic method are discussed in Section 4, followed by the conclusions in Section 5.

## 2 Improved Analysis of Alternating Stochastic Gradient Method

In this section, we will first provide background of bilevel problems and then introduce ALSET for stochastic nested problems.

### 2.1 Preliminaries

We use $\| \cdot \|$ to denote the $\ell_2$ norm for vectors and Frobenius norm for matrices. For convenience, we define the deterministic functions as $g(x, y) := \mathbb{E}_\phi[g(x, y; \phi)]$ and $f(x, y) := \mathbb{E}_\xi[f(x, y; \xi)]$.

We also define $\nabla^2_{yy} g(x, y)$ as the Hessian matrix of $g$ with respect to $y$ and define $\nabla^2_{xy} g(x, y)$ as

$$\nabla^2_{xy} g(x, y) := \begin{bmatrix} \frac{\partial^2}{\partial x_1 \partial y_1} g(x, y) & \cdots & \frac{\partial^2}{\partial x_1 \partial y_{d'}} g(x, y) \\ & \cdots & \\ \frac{\partial^2}{\partial x_d \partial y_1} g(x, y) & \cdots & \frac{\partial^2}{\partial x_d \partial y_{d'}} g(x, y) \end{bmatrix}.$$

We make the following assumptions, which are common in the bilevel optimization literature [16–18, 26].

**Assumption 1** (Lipschitz continuity). *Assume that $f, \nabla f, \nabla g, \nabla^2 g$ are respectively $\ell_{f,0}$, $\ell_{f,1}, \ell_{g,1}, \ell_{g,2}$-Lipschitz continuous; that is, for $z_1 := [x_1; y_1]$, $z_2 := [x_2; y_2]$, we have $\|f(x_1, y_1) - f(x_2, y_2)\| \leq \ell_{f,0}\|z_1 - z_2\|, \|\nabla f(x_1, y_1) - \nabla f(x_2, y_2)\| \leq \ell_{f,1}\|z_1 - z_2\|, \|\nabla g(x_1, y_1) - \nabla g(x_2, y_2)\| \leq \ell_{g,1}\|z_1 - z_2\|, \|\nabla^2 g(x_1, y_1) - \nabla^2 g(x_2, y_2)\| \leq \ell_{g,2}\|z_1 - z_2\|.$*

**Assumption 2** (Strong convexity of $g$ in $y$). *For any fixed $x$, $g(x, y)$ is $\mu_g$-strongly convex in $y$.*

Assumptions 1 and 2 together ensure that the first- and second-order derivations of $f(x, y), g(x, y)$, as well as the solution mapping $y^*(x)$, are well-behaved. Define the condition number $\kappa := \ell_{g,1}/\mu_g$.

**Assumption 3** (Stochastic derivatives). *The stochastic derivatives $\nabla f(x, y; \xi)$, $\nabla g(x, y; \phi)$, $\nabla^2 g(x, y, \phi)$ are unbiased estimators of $\nabla f(x, y)$, $\nabla g(x, y)$, $\nabla^2 g(x, y)$, respectively; and their variances are bounded by $\sigma_f^2, \sigma_{g,1}^2, \sigma_{g,2}^2$, respectively.*

|  | ALSET | SGDA | SMD |
|---|---|---|---|
| **batch size** | $\mathcal{O}(1)$ | $\mathcal{O}(\epsilon^{-1})$ | / |
| **$y$-update** | SGD | SGD | subproblem |
| **samples** | $\mathcal{O}(\kappa^3 \epsilon^{-2})$ | $\mathcal{O}(\kappa^3 \epsilon^{-2})$ | $\mathcal{O}(\kappa^3 \epsilon^{-2})$ |

Table 2: Sample complexity of stochastic min-max algorithms (BSA in [16], GDA in [28], SMD in [9]) to achieve an $\epsilon$-stationary point of $F(x)$.

|  | ALSET | SCGD | NASA |
|---|---|---|---|
| **batch size** | $\mathcal{O}(1)$ | $\mathcal{O}(1)$ | $\mathcal{O}(1)$ |
| **$y$-update** | SGD | SGD | correction |
| **samples** | $\mathcal{O}(\epsilon^{-2})$ | $\mathcal{O}(\epsilon^{-4})$ | $\mathcal{O}(\epsilon^{-2})$ |

Table 3: Sample complexity of stochastic compositional algorithms (SCGD in [12], NASA in [14]) to achieve an $\epsilon$-stationary point of $F(x)$.

Assumptions 2 and 3 together imply that the second moments are bounded by

$$\mathbb{E}_\xi[\|\nabla f(x,y;\xi)\|^2] \leq \ell_{f,0}^2 + \sigma_f^2 := C_f^2 \tag{5a}$$

$$\mathbb{E}_\phi[\|\nabla^2 g(x,y;\phi)\|^2] \leq \ell_{g,1}^2 + \sigma_{g,2}^2 := C_g^2. \tag{5b}$$

Assumption 3 is the counterpart of the unbiasedness and bounded variance assumption in the single-level stochastic optimization. In addition, the bounded moments in Assumption 3 ensure the Lipschitz continuity of the upper-level gradient $\nabla F(x)$.

We first highlight the inherent challenge of directly applying the alternating SGD method to the bilevel problem (1). To illustrate this point, we derive the gradient of the upper-level function $F(x)$ in the next proposition; see the proof in the supplementary document.

**Proposition 1.** *Under Assumptions 1–3, we have the gradients*

$$\nabla F(x) = \nabla_x f(x, y^*(x)) - \nabla_{xy}^2 g(x, y^*(x)) \big[\nabla_{yy}^2 g(x, y^*(x))\big]^{-1} \nabla_y f(x, y^*(x)). \tag{6}$$

*Furthermore, $\nabla F(x)$ and $y^*(x)$ are Lipschitz continuous with constants $L_F, L_y$, respectively.*

Notice that obtaining an unbiased stochastic estimate of $\nabla F(x)$ and applying SGD on $x$ face two main difficulties: i) the gradient $\nabla F(x)$ at $x$ depends on the minimizer of the lower-level problem $y^*(x)$; ii) even if $y^*(x)$ is known, it is hard to apply the stochastic approximation to obtain an unbiased estimate of $\nabla F(x)$ since $\nabla F(x)$ is nonlinear in $\nabla_{yy}^2 g(x, y^*(x))$.

Similar to some existing stochastic bilevel algorithms [16, 18, 17], we evaluate $\nabla F(x)$ on a certain vector $y$ in place of $y^*(x)$. Replacing the $y^*(x)$ in definition (6) by $y$, we define

$$\overline{\nabla}_x f(x, y) := \nabla_x f(x, y) - \nabla_{xy}^2 g(x, y) \big[\nabla_{yy}^2 g(x, y)\big]^{-1} \nabla_y f(x, y). \tag{7}$$

And to reduce the bias in (7), we estimate $\big[\nabla_{yy}^2 g(x, y)\big]^{-1}$ via

$$\big[\nabla_{yy}^2 g(x, y)\big]^{-1} \approx \Big[\frac{N}{\ell_{g,1}} \prod_{n=1}^{N'} \Big(I - \frac{1}{\ell_{g,1}} \nabla_{yy}^2 g(x, y; \phi_{(n)})\Big)\Big] \tag{8}$$

where $N'$ is drawn from $\{1, 2, \ldots, N\}$ uniformly at random and $\{\phi^{(1)}, \ldots, \phi^{(N')}\}$ are i.i.d. samples. It has been shown in [16] that using (8), the estimation bias of $\big[\nabla_{yy}^2 g(x, y)\big]^{-1}$ exponentially decreases with the number of samples $N$.

## 2.2  Main results: Tighter analysis of ALSET

In this subsection, we first describe the general ALSET algorithm for the stochastic bilevel problem, and then present its new convergence result.

This algorithm is very simple to implement. At each iteration $k$, ALSET alternates between the stochastic gradient update on $y^k$ and that on $x^k$. Although it is possible that $T = 1$, for generality, we run $T$ steps of SGD on

---

**Algorithm 1** ALSET for the stochastic bilevel problem (1)

1: **initialize:** $x^0, y^0$, stepsizes $\{\alpha_k, \beta_k\}$.
2: **for** $k = 0, 1, \ldots, K - 1$ **do**
3:     **for** $t = 0, 1, \ldots, T - 1$ **do**
4:         update $y^{k,t+1} = y^{k,t} - \beta_k h_g^{k,t}$   ▷ set $y^{k,0} = y^k$
5:     **end for**
6:     update $x^{k+1} = x^k - \alpha_k h_f^k$     ▷ set $y^{k+1} = y^{k,T}$
7: **end for**

---

the lower-level variable $y^k$ before updating upper-level variable $x^k$. With $\alpha_k$ and $\beta_k$ denoting the stepsizes of $x^k$ and $y^k$ that decrease at the same rate as SGD, the ALSET update is

$$y^{k,t+1} = y^{k,t} - \beta_k h_g^{k,t}, \ t = 0, \ldots, T \quad \text{with} \ y^{k,0} := y^k; \ y^{k+1} := y^{k,T} \tag{9a}$$

$$x^{k+1} = x^k - \alpha_k h_f^k \tag{9b}$$

where the update direction of $y$ is the stochastic gradient $h_g^{k,t} := \nabla_y g(x^k, y^{k,t}; \phi^{k,t})$; and, with the Hessian inverse estimator (8), the update direction of $x$ is the slightly biased gradient

$$h_f^k := \nabla_x f(x^k, y^{k+1}; \xi^k)$$

$$- \nabla_{xy}^2 g(x^k, y; \phi_{(0)}^k) \Big[\frac{N}{\ell_{g,1}} \prod_{n=1}^{N'} \Big(I - \frac{1}{\ell_{g,1}} \nabla_{yy}^2 g(x^k, y^{k+1}; \phi_{(n)}^k)\Big)\Big] \nabla_y f(x^k, y^{k+1}; \xi^k). \tag{10}$$

The alternating update (9) serves as a template for running SGD on stochastic nested problems. As we will show in the subsequent sections, we can generate stochastic algorithms for min-max, compositional, and even reinforcement learning problems following (9) as a template, but they differ in the particular forms of the stochastic gradients $h_g^k, h_f^k$ for the specific upper- and lower-level objective functions. See Algorithm 1 for a summary of ALSET for the bilevel problem.

**Comparison between ALSET with existing works.** Readers who are familiar with recent developments on stochastic optimization for bilevel problems may readily recognize the similarities between the general ALSET update (1) that we will analyze and the SGD-based updates in BSA [16], TTSA [18] and stocBiO [17]. However, the update (1) is different from BSA in that the number of $y$-update, denoted as $T$, is a constant in (1) that does not grow with the accuracy $\epsilon^{-1}$; the update (1) is different from stocBiO in that the stochastic gradient $h_g^{k,t}$ used in the $y$-update (9a) is obtained by a fixed batch size that does not depend on the accuracy $\epsilon^{-1}$; and, the update (1) is different from TTSA in that the stepsizes $\alpha_k$ and $\beta_k$ in (9) decrease at the same timescale.

We next present the convergence result of ALSET.

**Theorem 1** (Bilevel problems). *Suppose Assumptions 1–3 hold. Define the constants as*

$$\bar{\alpha}_1 = \frac{1}{2L_F + 4L_f L_y + \frac{2L_f L_{yx}}{L_y \eta}}, \quad \bar{\alpha}_2 = \frac{16T\mu_g \ell_{g,1}}{(\mu_g + \ell_{g,1})^2 (8L_f L_y + 2\eta L_{yx} \tilde{C}_f^2 \bar{\alpha}_1)} \tag{11}$$

*where $\eta > 0$ is a control constant that will be specified in each special case to achieve the best sample complexity. With $\alpha > 0$ being a control constant that will be specified later, choose the stepsizes as*

$$\alpha_k = \min\left\{\bar{\alpha}_1, \bar{\alpha}_2, \frac{\alpha}{\sqrt{K}}\right\} \quad \text{and} \quad \beta_k = \frac{8L_f L_y + 2\eta L_{yx} \tilde{C}_f^2 \bar{\alpha}_1}{4T\mu_g} \alpha_k. \tag{12}$$

*For any $T \geq 1$ and $N = \mathcal{O}(\log K)$, the iterates $\{x^k, y^k\}$ generated by Algorithm 1 satisfy*

$$\frac{1}{K} \sum_{k=1}^K \mathbb{E}\left[\left\|\nabla F(x^k)\right\|^2\right] = \mathcal{O}\left(\frac{1}{\sqrt{K}}\right) \quad \text{and} \quad \mathbb{E}\left[\left\|y^K - y^*(x^K)\right\|^2\right] = \mathcal{O}\left(\frac{1}{\sqrt{K}}\right) \tag{13}$$

*where $y^*(x^K)$ is the minimizer of the lower-level problem in (1b).*

**Proposition 2.** *Under the same assumptions and the choice of parameters of Theorem 1, with $\kappa := \frac{\ell_{g,1}}{\mu_g}$ being the condition number, select $\alpha = \Theta(\kappa^{-5/2})$, $T = \Theta(\kappa^4)$, $\eta = \mathcal{O}(\kappa)$ in (12), and then*

$$\frac{1}{K} \sum_{k=0}^{K-1} \mathbb{E}[\|\nabla F(x^k)\|^2] = \mathcal{O}\left(\frac{\kappa^3}{K} + \frac{\kappa^{\frac{5}{2}}}{\sqrt{K}}\right). \tag{14}$$

**Discussion of Theorem 1.** To achieve $\epsilon$-stationary point, we need $K = \mathcal{O}(\kappa^5 \epsilon^{-2})$, and the number of evaluations of $h_f^k, h_g^{k,t}$ are $\mathcal{O}(\kappa^5 \epsilon^{-2})$ and $\mathcal{O}(\kappa^9 \epsilon^{-2})$, respectively. Therefore, the sample complexity is on the same order of SGD's sample complexity for the single-level nonconvex problems [43], and improves the state-of-the-art single-loop TTSA's sample complexity $\mathcal{O}(\epsilon^{-5/2})$ [18]. Compared to [17], ALSET achieves the same sample complexity in terms of both $\epsilon$ and $\kappa$, without using a growing batch size. Importantly, we obtain this tighter bound without introducing additional assumptions.

## 2.3  Proof sketch

In this subsection, we highlight the key steps of the proof towards Theorem 1, and highlight the differences between our analysis and the existing ones.

For simplicity, we define the following Lyapunov function as $\mathbb{V}^k := F(x^k) + \frac{L_f}{L_y}\|y^k - y^*(x^k)\|^2$. We first quantify the difference between two Lyapunov functions as

$$\mathbb{V}^{k+1} - \mathbb{V}^k = \underbrace{F(x^{k+1}) - F(x^k)}_{\text{Lemma 1}} + \frac{L_f}{L_y} \underbrace{(\|y^{k+1} - y^*(x^{k+1})\|^2 - \|y^k - y^*(x^k)\|^2)}_{\text{Lemma 3}}. \tag{15}$$

The difference in (15) consists of two difference terms: the first term quantifies the descent of the overall objective functions; the second term characterizes the descent of the lower-level errors.

We will first analyze the descent of the upper-level objective in the next lemma.

**Lemma 1** (Descent of upper level). *Suppose Assumptions 1–3 hold. Define $\bar{h}_f^k := \mathbb{E}[h_f^k | x^k, y^{k+1}]$ and $\|\bar{h}_f^k - \overline{\nabla} f(x^k, y^{k+1})\| \le b_k$. The sequence of $x^k$ generated by Algorithm 1 satisfies*

$$\mathbb{E}[F(x^{k+1})] - \mathbb{E}[F(x^k)] \le -\frac{\alpha_k}{2}\mathbb{E}[\|\nabla F(x^k)\|^2] - \left(\frac{\alpha_k}{2} - \frac{L_F \alpha_k^2}{2}\right)\mathbb{E}[\|\bar{h}_f^k\|^2]$$

$$+ L_f^2 \alpha_k \mathbb{E}[\|y^{k+1} - y^*(x^k)\|^2] + \alpha_k b_k^2 + \frac{L_F \alpha_k^2}{2}\tilde{\sigma}_f^2 \qquad (16)$$

*where constants $L_f, L_F, \sigma_f^2$ are defined in Lemma 4 of the supplementary document.*

Lemma 1 implies that the descent of the upper-level objective functions depends on the error of the lower-level variable $y^k$. We will next analyze the error of the lower-level variable, which is the key step to improving the existing results.

Before we analyze the error of $y^k$, we introduce a lemma that characterizes the smoothness of $y^*(x)$ and the bounded moments of $h_f^k$. The smoothness and the bounded moments have not been explored by previous analysis such as [16–18], and they play an essential role in our improved analysis of $y^k$.

**Lemma 2** (Smoothness and boundedness). *Under Assumptions 1 and 2, we have*

$$\|\nabla y^*(x_1) - \nabla y^*(x_2)\| \le L_{yx}\|x_1 - x_2\|; \quad \mathbb{E}[\|h_f^k\|^2 | x^k, y^{k+1}] \le \tilde{C}_f^2 \qquad (17)$$

*where $L_{yx}$ and $\tilde{C}_f^2$ depend on the constants defined in Assumptions 1-2.*

Building upon Lemma 2, we establish the progress of the lower-level update.

**Lemma 3** (Error of lower level). *Suppose that Assumptions 1–3 hold, and $y^{k+1}$ is generated by running iteration (9) given $x^k$. If we choose $\beta_k \le \frac{2}{\mu_g + \ell_{g,1}}$, then $y^{k+1}$ satisfies*

$$\mathbb{E}[\|y^{k+1} - y^*(x^k)\|^2] \le (1 - \mu_g \beta_k)^T \mathbb{E}[\|y^k - y^*(x^k)\|^2] + T\beta_k^2 \sigma_{g,1}^2 \qquad (18a)$$

$$\mathbb{E}[\|y^{k+1} - y^*(x^{k+1})\|^2] \le \left(1 + 4L_f L_y \alpha_k + \frac{\eta L_{yx}\tilde{C}_f^2}{2}\alpha_k^2\right)\mathbb{E}[\|y^{k+1} - y^*(x^k)\|^2]$$

$$+ \left(L_y^2 + \frac{L_y}{4L_f \alpha_k} + \frac{L_{yx}}{2\eta}\right)\alpha_k^2 \mathbb{E}[\|\bar{h}_f^k\|^2] + \left(L_y^2 + \frac{L_{yx}}{2\eta}\right)\alpha_k^2 \tilde{\sigma}_f^2 \qquad (18b)$$

*where $\eta > 0$ is a fixed constant that will be chosen to obtain the tighter complexity bound.*

**The improved analysis of the lower-level problem.** Next we explain where we can obtain improved analysis. Plugging (18a) into (18b), and selecting stepsizes $\alpha_k, \beta_k$ properly, we can show that

$$\mathbb{E}[\|y^{k+1} - y^*(x^{k+1})\|^2] \le (1 - \delta_1)\mathbb{E}[\|y^k - y^*(x^k)\|^2] + \delta_2 \mathbb{E}[\|\bar{h}_f^k\|^2] + \delta_3 T\sigma_{g,1}^2 + \delta_4 \tilde{\sigma}_f^2 \qquad (19)$$

where the constants are $\delta_1 \in [0, 1), \delta_2 = \mathcal{O}(\alpha_k), \delta_3 = \mathcal{O}(\beta_k^2), \delta_4 = \mathcal{O}(\alpha_k^2)$. As we will show in our supplementary material, the term $\mathbb{E}[\|\bar{h}_f^k\|^2]$ will be canceled when combined with (16) in our analysis. Hence, choosing $\alpha_k = \mathcal{O}(k^{-1/2})$ and $\beta_k = \mathcal{O}(k^{-1/2})$ makes the variance terms in (19) decrease at the same $\mathcal{O}(k^{-1/2})$ rate as the vanilla SGD for stochastic non-nested problems.

As a comparison, the progress of the lower-level problem in [18, 17] can be summarized as

$$\mathbb{E}[\|y^{k+1} - y^*(x^{k+1})\|^2] \le (1 - \delta_1)\mathbb{E}[\|y^k - y^*(x^k)\|^2] + \delta_5 \sigma^2 \qquad (20)$$

where $\sigma^2$ is some variance term, and the constant is $\delta_5 = \mathcal{O}(\beta_k^2 + \alpha_k^2/\beta_k)$ or $\mathcal{O}(1/B_k)$ with $B_k$ being the batch size at iteration $k$. To balance the two terms in $\delta_5 = \mathcal{O}(\beta_k^2 + \alpha_k^2/\beta_k)$, two timescales of stepsizes $\lim_{k \to \infty} \alpha_k/\beta_k = 0$ are needed, which will make the variance term of the $y$-update in (20) and that of the $x$-update in (16) decrease at two different rates, slower than that of SGD; and to reduce $\delta_5 = \mathcal{O}(1/B_k)$, a growing batch size $B_k = \mathcal{O}(k)$ is needed for the $y$-update.

## 3 Applications to Stochastic Min-Max and Compositional Problems

Building upon the general results for the bilevel problems in Section 2, this section will identify special features of the stochastic min-max and stochastic compositional problems, and customize the general results to yield state-of-the-art convergence results for two special nested problems.

## 3.1 Stochastic min-max problems

We first apply our results to the stochastic min-max problem (2). In this special case, the lower-level function is $g(x, y; \phi) = -f(x, y; \xi)$, and the bilevel gradient in (6) reduces to

$$\nabla F(x) := \nabla_x f(x, y^*(x)) + \nabla_x y^*(x)^\top \nabla_y f(x, y^*(x)) = \nabla_x f(x, y^*(x)) \tag{21}$$

where the second equality follows from the optimality condition of the lower-level problem, i.e., $\nabla_y f(x, y^*(x)) = 0$. Similar to Section 2, we again approximate $\nabla F(x)$ on a certain vector $y$ in place of $y^*(x)$. Therefore, the alternating stochastic gradients for this special case are given by

$$h_g^{k,t} = -\nabla_y f(x^k, y^{k,t}; \xi_1^{k,t}) \quad \text{and} \quad h_f^k = \nabla_x f(x^k, y^{k+1}; \xi_2^k). \tag{22}$$

Plugging the stochastic gradient into the general update (9), we summarize the update in Algorithm 2. When the number of $y$-update is $T = 1$, the ALSET algorithm reduces to the SGDA method in [28].

**Proposition 3** (Min-max problems)**.** *Choose the same choice of parameters as those in Theorem 1, and follow the same assumption as those in Theorem 1 except that $f(\cdot, y)$ is only Lipchitz over $x \in \mathbb{R}^d$ but not that $f(x, \cdot)$ is Lipschitz continuous over $y \in \mathbb{R}^{d'}$. If we select $\alpha = \Theta(\kappa^{-1})$, $T = \Theta(\kappa)$, $\eta = 1$ in (12), the iterates generated by Algorithm 2 satisfy*

$$\frac{1}{K} \sum_{k=0}^{K-1} \mathbb{E}\left[\left\|\nabla F(x^k)\right\|^2\right] = \mathcal{O}\left(\frac{\kappa^2}{K} + \frac{\kappa}{\sqrt{K}}\right). \tag{23}$$

Proposition 3 implies that for the min-max problem, the convergence rate of ALSET to the stationary point of $F(x) := \max_{y \in \mathbb{R}^{d'}} \mathbb{E}_\xi [f(x, y; \xi)]$ is $\mathcal{O}(K^{-1/2})$. To achieve $\epsilon$-stationary point, we need $K = \mathcal{O}(\kappa^2 \epsilon^{-2})$. And the number of gradient evaluations for $h_f^k, h_g^{k,t}$ are $\mathcal{O}(\kappa^2 \epsilon^{-2})$ and $\mathcal{O}(\kappa^3 \epsilon^{-2})$, respectively. Comparing with the results in [28], we achieve the same sample complexity without an increasing batch size $\mathcal{O}(\epsilon^{-1})$, and improve their sample complexity $\mathcal{O}(\epsilon^{-5/2})$ under a fixed batch size.

---

**Algorithm 2** ALSET for the min-max problem (2)

1: **initialize:** $x^0, y^0$, stepsizes $\{\alpha_k, \beta_k\}$.
2: **for** $k = 0, 1, \ldots, K - 1$ **do**
3:      set $y^{k,0} = y^k$
4:      **for** $t = 0, 1, \ldots, T - 1$ **do**
5:          update $y^{k,t+1} = y^{k,t} - \beta_k \nabla_y f(x^k, y^{k,t}; \xi_1^{k,t})$
6:      **end for**
7:      set $y^{k+1} = y^{k,T}$
8:      update $x^{k+1} = x^k - \alpha_k \nabla_x f(x^k, y^{k+1}; \xi_2^k)$
9: **end for**

---

However, it is also worth mentioning that compared with [28], our analysis requires the additional Lipschitz continuity assumption of $f(\cdot, y)$ over $x \in \mathbb{R}^d$, which inherits from the analysis for the general bilevel problem. Therefore, our result complements, rather than improves, the analysis in [28]. We view our contribution in min-max problems as a supplementary of existing results.

## 3.2 Stochastic compositional problems

In this section, we apply our results to the stochastic compositional problem (3). In this special case, the upper-level function is $f(x, y; \xi) := f(y; \xi)$, and the lower-level function is $g(x, y; \phi) = \|y - h(x; \phi)\|^2$, and the bilevel gradient in (6) reduces to

$$\nabla F(x) := \nabla_x f(x, y^*(x)) - \nabla_{xy}^2 g(x, y^*(x)) \left[\nabla_{yy}^2 g(x, y^*(x))\right]^{-1} \nabla_y f(x, y^*(x))$$
$$= \nabla h(x; \phi)^\top \nabla_y f(y^*(x)) \tag{24}$$

where we use the fact that $\nabla_{yy}^2 g(x, y; \phi) = \mathbf{I}_{d' \times d'}, \nabla_{xy}^2 g(x, y; \phi) = -\nabla h(x; \phi)^\top$. Similar to Section 2, we again evaluate $\nabla F(x)$ on a certain vector $y$ in place of $y^*(x)$. Therefore, by choosing $T = 1$, the alternating stochastic gradients $h_f^k, h_g^{k,t}$ for this special case are much simpler, given by

$$h_g^{k,t} = h_g^k = y^k - h(x^k; \phi^k) \quad \text{and} \quad h_f^k = \nabla h(x^k; \phi^k) \nabla f(y^{k+1}; \xi^k). \tag{25}$$

Plugging the stochastic gradient into the general update (9), we summarize the update in Algorithm 3. When $T = 1$, the ALSET algorithm reduces to SCGD proposed in [12].

In the supplementary document, we have verified that the standard assumptions of stochastic compositional optimization in [12, 37, 14, 41, 38] are sufficient for Assumptions 1–3 to hold.

**Proposition 4** (Compositional problems). *Under the same assumptions and the parameters as those in Theorem 1, if we select $T = 1, \alpha = 1, \eta = \frac{1}{L_{yx}}$ in (12), the iterates of Algorithm 3 satisfy*

$$\frac{1}{K} \sum_{k=1}^{K} \mathbb{E}\left[\left\|\nabla F(x^k)\right\|^2\right] = \mathcal{O}\left(\frac{1}{\sqrt{K}}\right). \quad (26)$$

Since each iteration of ALSET only uses $\mathcal{O}(1)$ samples (see Algorithm 3), Proposition 4 implies that the sample complexity to achieve an $\epsilon$-stationary point of (3) is $\mathcal{O}(\epsilon^{-2})$. Comparing with the results

---

**Algorithm 3** ALSET for the compositional problem (3)

---
1: **initialize:** $x^0, y^0$, stepsizes $\{\alpha_k, \beta_k\}$.
2: **for** $k = 0, 1, \ldots, K - 1$ **do**
3:     update $y^{k+1} = y^k - \beta_k(y^k - h(x^k; \phi^k))$
4:     update $x^{k+1} = x^k - \alpha_k \nabla f(y^{k+1}; \xi^k) \nabla h(x^k; \phi^k)$
5: **end for**

---

of the SCGD method in [12], our result improves the sample complexity $\mathcal{O}(\epsilon^{-4})$ under a fixed batch size. Importantly, our analysis does not introduce additional assumption compared to [12].

## 4 Applications to Actor-Critic Methods

In this section, we apply our tighter analysis to the actor-critic (AC) method with linear value function approximation [44], which can be viewed as a special case of the stochastic bilevel algorithm [45, 46].

Consider a Markov decision process described by $\mathcal{M} = \{\mathcal{S}, \mathcal{A}, \mathcal{P}, R, \gamma\}$, where $\mathcal{S}$ is the state space, $\mathcal{A}$ is the action space, $\mathcal{P}(s'|s, a)$ is the probability of transitioning to $s' \in \mathcal{S}$ given state $s \in \mathcal{S}$ and action $a \in \mathcal{A}$, and $R(s, a, s')$ is the reward associated with $(s, a, s')$, and $\gamma \in [0, 1)$ is a discount factor. For a policy $\pi_\theta$, define the value function $V_{\pi_\theta}(s)$ that satisfies the Bellman equation [47]

$$V_{\pi_\theta}(s) = \mathbb{E}_{a \sim \pi_\theta(\cdot|s), s' \sim \mathcal{P}(\cdot|s,a)} \left[r(s, a, s') + \gamma V_{\pi_\theta}(s')\right]. \quad (27)$$

Given the state feature mapping $\phi(\cdot) : \mathcal{S} \to \mathbb{R}^{d_y}$, we approximate the value function linearly as $V_{\pi_\theta}(s) \approx \hat{V}_y(s) := \phi(s)^\top y$, where $y \in \mathbb{R}^{d_y}$ is the critic parameter. The task of finding the best $y$ such that $V_{\pi_\theta}(s) \approx \hat{V}_y(s)$ is usually addressed by TD learning [48].

Defining the stationary distribution induced by the policy parameter $\theta_k$ as $\mu_{\theta_k}$ and the $k$th transition as $\xi_k := (s_k, a_k, s_{k+1})$, which is sampled from $s_k \sim \mu_{\theta_k}, a \sim \pi_{\theta_k}, s_{k+1} \sim \mathcal{P}$, the TD-error is

$$\hat{\delta}(\xi_k, y_k) := r(s_k, a_k, s_{k+1}) + \gamma \phi(s_{k+1})^\top y_k - \phi(s_k)^\top y_k \quad (28)$$

and the critic gradient $h_g(\xi_k, y_k) := \hat{\delta}(\xi_k, y_k) \nabla \hat{V}_{y_k}(s_k)$. We update the parameter $y$ via

$$y_{k+1} = \Pi_{R_y}\left(y_k + \beta_k h_g(\xi_k, y_k)\right), \quad (29)$$

where $\beta_k$ is the critic stepsize, and $\Pi_{R_y}$ is the projection to control the norm of the gradient. A pre-defined constant $R_y$ will be specified in the supplementary document.

The goal of policy optimization is to solve $\max_{\theta \in \mathbb{R}^d} F(\theta)$ with $F(\theta) := \mathbb{E}_{s \sim \eta}[V_{\pi_\theta}(s)]$, where $\eta$ is the initial distribution. Leveraging the value function approximation and the policy gradient theorem [49], we have the policy gradient $h_f(\xi, \theta, y) := \hat{\delta}(\xi, y) \psi_\theta(s, a)$, which gives the policy update

$$\theta_{k+1} = \theta_k + \alpha_k h_f(\xi'_k, \theta_k, y_{k+1}), \quad (30)$$

where $\alpha_k$ is the stepsize and $\psi_\theta(s, a) := \nabla \log \pi_\theta(a|s)$. Note that the sample $\xi'_k := (s'_k, a'_k, s'_{k+1})$ used in (30) is independent from $\xi_k$ in (29). Specifically, $\xi'_k$ is sampled from $s'_k \sim d_{\theta_k}, a'_k \sim \pi_{\theta_k}, s'_{k+1} \sim \mathcal{P}$ with $d_{\theta_k}$ being the discounted state action visitation measure under $\theta_k$.

The alternating AC update (29)-(30) is a special case of ALSET, where the critic update is the lower-level update, and the actor update is the upper-level update.

Due to space limitation, we will directly present the results of the alternating AC next, and defer presentation of the proof and the corresponding assumptions, which are the counterparts of Assumptions 1–3 in the context of AC, to the supplementary document.

**Theorem 2** (Actor-critic). *Under the some regularity conditions that are specified in the supplementary document, selecting step size $\alpha_k = \alpha = \mathcal{O}(\frac{1}{\sqrt{K}})$, $\beta_k = \beta = \mathcal{O}(\frac{1}{\sqrt{K}})$, it holds*

$$\frac{1}{K} \sum_{k=1}^{K} \mathbb{E}\left[\|\nabla F(\theta_k)\|^2\right] = \mathcal{O}\left(\frac{1}{\sqrt{K}}\right) + \epsilon_{\text{app}} \quad (31)$$

*where $\epsilon_{\text{app}}$, defined in the supplementary document, captures the richness of the linear function class.*

**Both sides of Theorem 2.** As an application of our tighter analysis, Theorem 2 establishes for the first time that the sample complexity of the single-loop alternating actor-critic method is $\mathcal{O}(\epsilon^{-2})$. On the positive side, this new result improves the previous complexity $\mathcal{O}(\epsilon^{-5/2})$ for the single-loop AC [50], and $\mathcal{O}(\epsilon^{-2} \log \epsilon^{-1})$ for the nested-loop AC [51], and matches $\mathcal{O}(\epsilon^{-2})$ for AC with an exact critic oracle [52]. In addition to using two independent samples, one limitation of our result is that inheriting from the analysis for the general bilevel case, our analysis of AC requires the smoothness of the critic fixed-point $y^*(\theta)$. As shown in the supplementary document, this implicitly requires the additional bounded and Lipschitz continuity assumption on the stationary distribution $\mu_\theta$. The removal of this assumption and the extension to Markovian sampling are left for future research.

## 5   Preliminary Experiments

To validate our new theoretical results, we have conducted the simple experiment using the risk-averse portfolio management task on a benchmark dataset - 100 Book-to-Market. This is a typical application of stochastic compositional optimization (3) that is used in [40, 41]. We compared the popular **two-timescale** SCGD approach [12] with our **single-timescale** ALSET approach.

We use the same initialization of $x^0, y^0$ for both SCGD and ALSET, and tune the stepsizes $\alpha_k, \beta_k$ by following the suggested order in the original SCGD paper and then using a grid search for the multiplicative constant $c$, that is

SCGD: $\alpha_k = c\,k^{-3/4}$, $\beta_k = c\,k^{-1/2}$;
ALSET: $\alpha_k = c\,k^{-1/2}$, $\beta_k = c\,k^{-1/2}$.

| Iter $k$ | $\ln k$ | SCGD | ALSET | ALSET-const |
|---|---|---|---|---|
| 10 | 2.30 | 5.32 | 5.31 | 5.63 |
| 100 | 4.61 | 3.78 | 3.49 | 3.63 |
| 200 | 5.30 | 3.40 | 2.94 | 3.06 |
| 400 | 5.99 | 3.04 | 2.40 | 2.55 |
| 1000 | 6.91 | 2.57 | 1.65 | 2.06 |

Table 4: Comparison of $\ln(\frac{1}{K} \sum_{k=1}^{K} \|\nabla F(x^k)\|^2)$ among the two-timescale and single-timescale algorithms.

The constant $c$ is chosen from the searching grid $\{10^{-3}, 5 \times 10^{-4}, 10^{-4}\}$ and is optimized for each algorithm in terms of ergodic average gradient norm versus the number of iterations. In Table 4, we report the logarithmic value of the average gradient norm performance of SCGD, ALSET with both the above decreasing stepsizes and ALSET-const with the constant stepsizes (replacing $k$ with $K = 1000$). Since SCGD and ALSET use the same number of samples and gradient evaluations per iteration, we report the progress in terms of iterations. By calculating the decay rate, we can observe that the empirical convergence rate of ALSET is no worse than the theoretical rate $\mathcal{O}(k^{-1/2})$, and ALSET outperforms SCGD thanks to its single-timescale stepsizes. We will pursue more comprehensive experiments in our future work.

## 6   Conclusions

This paper unifies several SGD-type updates for stochastic nested problems into a single nested SGD approach that we term ALternating Stochastic gradient dEscenT (ALSET) method. ALSET runs in the single-timescale and uses a fixed batch size. This paper presents a tighter analysis for using ALSET to solve stochastic nested problems. Under the new analysis, to achieve an $\epsilon$-stationary point of the nested problem, ALSET requires $\mathcal{O}(\epsilon^{-2})$ samples in total. As a by-product, this general result also improves the existing sample complexity of the min-max and compositional cases. It matches the sample complexity of SGD for single-level stochastic problems. Applying our analysis to an alternating version of the actor-critic algorithm also yields a state-of-the-art sample complexity.

Potential limitations of our results include additional assumptions in the min-max and actor-critic cases, which inherit from the assumptions of general bilevel problems. Nevertheless, our work can also lead to promising future research in understanding the theoretical performance of many successful empirical nested optimization algorithms. To this end, our future work consists of relaxing the regularity conditions needed to achieve our theoretical results and Possible extensions include applying our the tighter analysis in this paper to the existing two-timescale Hessian-free bilevel optimization algorithms and decentralized stochastic nested optimization algorithms.

## Acknowledgements

The work of T. Chen was partially supported by NSF Grant 2047177 and the RPI-IBM Artificial Intelligence Research Collaboration (AIRC). The work of Y. Sun was partially supported by ONR

Grant N000141712162 and AFOSR MURI FA9550-18-1-0502. We thank anonymous reviewers for their valuable feedback on improving the current paper.

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
