# Supplementary Material for "Closing the Gap: Tighter Analysis of Alternating Stochastic Gradient Methods for Bilevel Problems"

## Table of Contents

## A  Proof for stochastic bilevel problem

### A.1  Auxiliary Lemmas

Throughout the proof, we use $\mathcal{F}_{k,t} = \sigma\{y^0, x^0, \ldots, y^k, x^k, y^{k,1}, \ldots, y^{k,t}\}$, $\mathcal{F}'_k = \sigma\{y^0, x^0, \ldots, y^{k+1}\}$, where $\sigma\{\cdot\}$ denotes the $\sigma$-algebra generated by the random variables.

We first present some results that will be used frequently in the proof.

**Proposition 5** (Restatement of Proposition 1). *Under Assumptions 1–3, we have the gradients*

$$\nabla F(x) = \nabla_x f(x, y^*(x)) - \nabla^2_{xy} g(x, y^*(x)) \left[\nabla^2_{yy} g(x, y^*(x))\right]^{-1} \nabla_y f(x, y^*(x)). \quad (32)$$

*Proof.* Define the Jacobian matrix

$$\nabla_x y(x) = \begin{bmatrix} \frac{\partial}{\partial x_1} y_1(x) & \cdots & \frac{\partial}{\partial x_d} y_1(x) \\ & \cdots & \\ \frac{\partial}{\partial x_1} y_{d'}(x) & \cdots & \frac{\partial}{\partial x_d} y_{d'}(x) \end{bmatrix}.$$

By the chain rule, it follows that

$$\nabla F(x) := \nabla_x f(x, y^*(x)) + \nabla_x y^*(x)^\top \nabla_y f(x, y^*(x)). \quad (33)$$

The minimizer $y^*(x)$ satisfies

$$\nabla_y g(x, y^*(x)) = 0, \quad \text{thus} \quad \nabla_x(\nabla_y g(x, y^*(x))) = 0, \quad (34)$$

from which and the chain rule, it follows that

$$\nabla^2_{xy} g(x, y^*(x)) + \nabla_x y^*(x)^\top \nabla^2_{yy} g(x, y^*(x)) = 0.$$

By Assumption 2, $\nabla^2_{yy} g(x, y^*(x))$ is invertible, so from the last equation,

$$\nabla_x y^*(x)^\top := -\nabla^2_{xy} g(x, y^*(x)) \left[\nabla^2_{yy} g(x, y^*(x))\right]^{-1}. \quad (35)$$

Substituting (35) into (33) yields (6).

**Lemma 4** ([16, Lemma 2.2]). *Under Assumptions 1 and 2, we have*

$$\|\overline{\nabla}_x f(x, y^*(x)) - \overline{\nabla}_x f(x, y)\| \le L_f \|y^*(x) - y\| \tag{36a}$$
$$\|\nabla F(x_1) - \nabla F(x_2)\| \le L_F \|x_1 - x_2\| \tag{36b}$$
$$\|y^*(x_1) - y^*(x_2)\| \le L_y \|x_1 - x_2\| \tag{36c}$$

*with the constants $L_f, L_y, L_F$ given by*

$$L_f := \ell_{f,1} + \frac{\ell_{g,1}\ell_{f,1}}{\mu_g} + \frac{\ell_{f,0}}{\mu_g}\left(\ell_{g,2} + \frac{\ell_{g,1}\ell_{g,2}}{\mu_g}\right) = \mathcal{O}(\kappa^2), \quad L_y := \frac{\ell_{g,1}}{\mu_g} = \mathcal{O}(\kappa)$$

$$L_F := \ell_{f,1} + \frac{\ell_{g,1}(\ell_{f,1} + L_f)}{\mu_g} + \frac{\ell_{f,0}}{\mu_g}\left(\ell_{g,2} + \frac{\ell_{g,1}\ell_{g,2}}{\mu_g}\right) = \mathcal{O}(\kappa^3),$$

*where the other constants are defined in Assumptions 1–3.*

**Lemma 5** ([18, Lemma 11]). *Recall the definition of $h_f^k$ in (10). Define*

$$\bar{h}_f^k := \mathbb{E}[h_f^k | \mathcal{F}_k'].$$

*We have*

$$\|\overline{\nabla}_x f(x^k, y^{k+1}) - \bar{h}_f^k\| \le \ell_{g,1}\ell_{f,1}\frac{1}{\mu_g}\left(1 - \frac{\mu_g}{\ell_{g,1}}\right)^N =: b_k$$

$$\mathbb{E}[\|h_f^k - \bar{h}_f^k\|^2] \le \sigma_f^2 + \frac{3}{\mu_g^2}\left[(\sigma_f^2 + \ell_{f,0}^2)(\sigma_{g,2}^2 + 2\ell_{g,1}^2) + \sigma_f^2\ell_{g,1}^2\right] =: \tilde{\sigma}_f^2 = \mathcal{O}(\kappa^2),$$

*where $\kappa$ is the condition number defined below Assumption 2.*

## A.2  Proof of Lemma 1

Using the Lipschitz property of $\nabla F$ in Lemma 4, we have

$$\mathbb{E}[F(x^{k+1})|\mathcal{F}_k'] \le F(x^k) + \mathbb{E}[\langle \nabla F(x^k), x^{k+1} - x^k\rangle|\mathcal{F}_k'] + \frac{L_F}{2}\mathbb{E}[\|x^{k+1} - x^k\|^2|\mathcal{F}_k']$$

$$= F(x^k) - \alpha_k\langle \nabla F(x^k), \bar{h}_f^k\rangle + \frac{L_F\alpha_k^2}{2}\mathbb{E}[\|h_f^k\|^2|\mathcal{F}_k']$$

$$\stackrel{(a)}{=} F(x^k) - \frac{\alpha_k}{2}\|\nabla F(x^k)\|^2 - \frac{\alpha_k}{2}\|\bar{h}_f^k\|^2 + \frac{\alpha_k}{2}\|\nabla F(x^k) - \bar{h}_f^k\|^2$$

$$+ \frac{L_F\alpha_k^2}{2}\|\bar{h}_f^k\|^2 + \frac{L_F\alpha_k^2}{2}\mathbb{E}[\|h_f^k - \bar{h}_f^k\|^2|\mathcal{F}_k']$$

$$\stackrel{(b)}{\le} F(x^k) - \frac{\alpha_k}{2}\|\nabla F(x^k)\|^2 - \left(\frac{\alpha_k}{2} - \frac{L_F\alpha_k^2}{2}\right)\|\bar{h}_f^k\|^2$$

$$+ \frac{\alpha_k}{2}\|\nabla F(x^k) - \bar{h}_f^k\|^2 + \frac{L_F\alpha_k^2}{2}\tilde{\sigma}_f^2 \tag{37}$$

where (a) uses $2a^\top b = \|a\|^2 + \|b\|^2 - \|a - b\|^2$ twice and (b) uses Lemma 5.

We decompose the gradient bias term as follows

$$\|\nabla F(x^k) - \bar{h}_f^k\|^2 = \|\overline{\nabla}f(x^k, y^*(x^k)) - \overline{\nabla}f(x^k, y^{k+1}) + \overline{\nabla}f(x^k, y^{k+1}) - \bar{h}_f^k\|^2$$

$$\le 2\|\overline{\nabla}f(x^k, y^*(x^k)) - \overline{\nabla}f(x^k, y^{k+1})\|^2 + 2\|\overline{\nabla}f(x^k, y^{k+1}) - \bar{h}_f^k\|^2$$

$$\stackrel{(a)}{\le} 2L_f^2\|y^{k+1} - y^*(x^k)\|^2 + 2b_k^2 \tag{38}$$

where (a) follows from Lemma 4 and Lemma 5. Plugging (38) into (37) completes the proof.

## A.3 Proof of Lemma 2

Recalling the definition of $\nabla_x y^*(x)$ in (35), for any $x_1, x_2$, we have

$$\|\nabla_x y^*(x_1) - \nabla_x y^*(x_2)\| \tag{39}$$
$$= \|\nabla_{xy}^2 g(x_1, y^*(x_1))[\nabla_{yy}^2 g(x_1, y^*(x_1))]^{-1} - \nabla_{xy}^2 g(x_2, y^*(x_2))[\nabla_{yy}^2 g(x_2, y^*(x_2))]^{-1}\|$$
$$\leq \|\nabla_{xy}^2 g(x_1, y^*(x_1)) - \nabla_{xy}^2 g(x_2, y^*(x_2))\|\|[\nabla_{yy}^2 g(x_1, y^*(x_1))]^{-1}\|$$
$$\quad + \|\nabla_{xy}^2 g(x_2, y^*(x_2))\|\|[\nabla_{yy}^2 g(x_1, y^*(x_1))]^{-1} - [\nabla_{yy}^2 g(x_2, y^*(x_2))]^{-1}\|$$
$$\overset{(a)}{\leq} \frac{1}{\mu_g}\|\nabla_{xy}^2 g(x_1, y^*(x_1)) - \nabla_{xy}^2 g(x_2, y^*(x_2))\|$$
$$\quad + \ell_{g,1}\|[\nabla_{yy}^2 g(x_1, y^*(x_1))]^{-1}\left(\nabla_{yy}^2 g(x_1, y^*(x_1)) - \nabla_{yy}^2 g(x_2, y^*(x_2))\right)[\nabla_{yy}^2 g(x_2, y^*(x_2))]^{-1}\|$$
$$\overset{(b)}{\leq} \frac{1}{\mu_g}\|\nabla_{xy}^2 g(x_1, y^*(x_1)) - \nabla_{xy}^2 g(x_2, y^*(x_2))\| + \frac{\ell_{g,1}}{\mu_g^2}\|\nabla_{yy}^2 g(x_1, y^*(x_1)) - \nabla_{yy}^2 g(x_2, y^*(x_2))\|$$

where both (a) and (b) follow from Assumption 1 and 2.

In addition, we have that

$$\frac{1}{\mu_g}\|\nabla_{xy}^2 g(x_1, y^*(x_1)) - \nabla_{xy}^2 g(x_2, y^*(x_2))\| + \frac{\ell_{g,1}}{\mu_g^2}\|\nabla_{yy}^2 g(x_1, y^*(x_1)) - \nabla_{yy}^2 g(x_2, y^*(x_2))\|$$
$$\leq \frac{\ell_{g,2}}{\mu_g}\|x_1 - x_2\| + \frac{\ell_{g,2}}{\mu_g}\|y^*(x_1) - y^*(x_2)\| + \frac{\ell_{g,1}\ell_{g,2}}{\mu_g^2}\|x_1 - x_2\| + \frac{\ell_{g,1}\ell_{g,2}}{\mu_g^2}\|y^*(x_1) - y^*(x_2)\|$$
$$\overset{(c)}{\leq} \left(\frac{\ell_{g,2} + \ell_{g,2}L_y}{\mu_g} + \frac{\ell_{g,1}(\ell_{g,2} + \ell_{g,2}L_y)}{\mu_g^2}\right)\|x_1 - x_2\| \tag{40}$$

where (c) follows from Lemma 4.

Next we derive the bound of $h_f^k$,

$$\mathbb{E}[\|h_f^k\|^2|\mathcal{F}_k'] = \|\bar{h}_f^k\|^2 + \mathbb{E}[\|h_f^k - \bar{h}_f^k\|^2|\mathcal{F}_k']$$
$$\overset{(d)}{\leq} (\|\overline{\nabla}f(x^k, y^{k+1})\| + \|\bar{h}_f^k - \overline{\nabla}f(x^k, y^{k+1})\|)^2 + \tilde{\sigma}_f^2$$
$$\overset{(e)}{\leq} \left(\ell_{f,0} + \frac{\ell_{f,0}\ell_{g,1}}{\mu_g} + \frac{\ell_{g,1}\ell_{f,1}}{\mu_g}\left(1 - \frac{\mu_g}{\ell_{g,1}}\right)^N\right)^2 + \tilde{\sigma}_f^2$$
$$\leq \left(\ell_{f,0} + \frac{\ell_{f,0}\ell_{g,1}}{\mu_g} + \frac{\ell_{g,1}\ell_{f,1}}{\mu_g}\right)^2 + \tilde{\sigma}_f^2 \tag{41}$$

where (d) is from Lemma 5, and (e) is due to

$$\|\overline{\nabla}_x f(x, y)\| = \|\nabla_x f(x, y) - \nabla_{xy}^2 g(x, y)\left[\nabla_{yy}^2 g(x, y)\right]^{-1}\nabla_y f(x, y)\|$$
$$\leq \|\nabla_x f(x, y)\| + \|\nabla_{xy}^2 g(x, y)\|\left\|\left[\nabla_{yy}^2 g(x, y)\right]^{-1}\right\|\|\nabla_y f(x, y)\|$$
$$\leq \ell_{f,0} + \ell_{g,1}\frac{1}{\mu_g}\ell_{f,0}.$$

As a result, we have

$$L_{yx} := \frac{\ell_{g,2} + \ell_{g,2}L_y}{\mu_g} + \frac{\ell_{g,1}(\ell_{g,2} + \ell_{g,2}L_y)}{\mu_g^2} = \mathcal{O}(\kappa^3) \tag{42}$$

$$\tilde{C}_f^2 := \left(l_{f,0} + \frac{\ell_{g,1}}{\mu_g}\ell_{f,1} + \ell_{g,1}\ell_{f,1}\frac{1}{\mu_g}\right)^2 + \tilde{\sigma}_f^2 = \mathcal{O}(\kappa^2) \tag{43}$$

from which the proof is complete.

## A.4 Proof of Lemma 3

This part of analysis is very important to obtain our improved results. We start by decomposing the error of the lower level variable as

$$\|y^{k+1} - y^*(x^{k+1})\|^2 = \underbrace{\|y^{k+1} - y^*(x^k)\|^2}_{J_1} + \underbrace{\|y^*(x^{k+1}) - y^*(x^k)\|^2}_{J_2}$$
$$+ 2\underbrace{\langle y^{k+1} - y^*(x^k), y^*(x^k) - y^*(x^{k+1})\rangle}_{J_3}. \tag{44}$$

Notice that $y^{k+1} = y^{k,T}$ as defined in (9a). We first analyze

$$\mathbb{E}[\|y^{k,t+1} - y^*(x^k)\|^2|\mathcal{F}_k^t]$$
$$= \mathbb{E}[\|y^{k,t} - \beta_k h_g^{k,t} - y^*(x^k)\|^2|\mathcal{F}_k^t]$$
$$= \|y^{k,t} - y^*(x^k)\|^2 - 2\beta_k\langle y^{k,t} - y^*(x^k), \mathbb{E}[h_g^{k,t}|\mathcal{F}_k^t]\rangle + \beta_k^2\mathbb{E}[\|h_g^{k,t}\|^2|\mathcal{F}_k^t]$$
$$\overset{(a)}{\leq} \|y^{k,t} - y^*(x^k)\|^2 - 2\beta_k\langle y^{k,t} - y^*(x^k), \nabla g(x^k, y^{k,t})\rangle + \beta_k^2\|\nabla g(x^k, y^{k,t})\|^2 + \beta_k^2\sigma_{g,1}^2$$
$$\overset{(b)}{\leq} \left(1 - \frac{2\mu_g\ell_{g,1}}{\mu_g + \ell_{g,1}}\beta_k\right)\|y^{k,t} - y^*(x^k)\|^2 + \beta_k\left(\beta_k - \frac{2}{\mu_g + \ell_{g,1}}\right)\|\nabla_y g(x^k, y^{k,t})\|^2 + \beta_k^2\sigma_{g,1}^2$$
$$\overset{(c)}{\leq} (1 - \rho_g\beta_k)\|y^{k,t} - y^*(x^k)\|^2 + \beta_k^2\sigma_{g,1}^2 \tag{45}$$

where (a) comes from the fact that $\text{Var}[X] = \mathbb{E}[X^2] - \mathbb{E}[X]^2$, (b) follows from the $\mu_g$-strong convexity and $\ell_{g,1}$ smoothness of $g(x, y)$ [53, Theorem 2.1.11], and (c) follows from the choice of stepsize $\beta_k \leq \frac{2}{\mu_g+\ell_{g,1}}$ in (12) and the definition of $\rho_g := \frac{2\mu_g\ell_{g,1}}{\mu_g+\ell_{g,1}}$.

Taking expectation over $\mathcal{F}_k^t$ on both sides of (45) and using induction, we are able to get

$$\mathbb{E}[J_1] = \mathbb{E}[\|y^{k+1} - y^*(x^k)\|^2] \leq (1 - \rho_g\beta_k)^T\mathbb{E}[\|y^k - y^*(x^k)\|^2] + T\beta_k^2\sigma_{g,1}^2. \tag{46}$$

The upper bound of $J_2$ can be derived as

$$\mathbb{E}[J_2] = \mathbb{E}[\|y^*(x^{k+1}) - y^*(x^k)\|^2] \leq L_y^2\mathbb{E}[\|x^{k+1} - x^k\|^2]$$
$$= L_y^2\alpha_k^2\mathbb{E}\left[\mathbb{E}[\|h_f^k - \bar{h}_f^k + \bar{h}_f^k\|^2|\mathcal{F}_k']\right]$$
$$\leq L_y^2\alpha_k^2(\mathbb{E}[\|\bar{h}_f^k\|^2] + \tilde{\sigma}_f^2) \tag{47}$$

where the inequality follows from Lemma 5.

Our analysis of the term $J_3$ is very different from existing bilevel optimization literature [16, 18, 17, 25, 26]. The term $J_3$ can be decomposed as

$$\mathbb{E}[J_3] = \underbrace{-\mathbb{E}[\langle y^{k+1} - y^*(x^k), \nabla y^*(x^k)(x^{k+1} - x^k)\rangle]}_{J_3^1}$$
$$\underbrace{-\mathbb{E}[\langle y^{k+1} - y^*(x^k), y^*(x^{k+1}) - y^*(x^k) - \nabla y^*(x^k)(x^{k+1} - x^k)\rangle]}_{J_3^2}. \tag{48}$$

Using the alternating update of $x$ and $y$, e.g., $x^k \to y^{k+1} \to x^{k+1}$, we can bound $J_3^1$ by

$$-\mathbb{E}[\langle y^{k+1} - y^*(x^k), \nabla y^*(x^k)(x^{k+1} - x^k)\rangle] = -\mathbb{E}[\langle y^{k+1} - y^*(x^k), \mathbb{E}[\nabla y^*(x^k)(x^{k+1} - x^k) \mid \mathcal{F}_k']\rangle]$$
$$\overset{(d)}{=} -\alpha_k\mathbb{E}[\langle y^{k+1} - y^*(x^k), \nabla y^*(x^k)\bar{h}_f^k\rangle]$$
$$\leq \alpha_k\mathbb{E}[\|y^{k+1} - y^*(x^k)\|\|\nabla y^*(x^k)\bar{h}_f^k\|]$$
$$\overset{(e)}{\leq} \alpha_k L_y\mathbb{E}[\|y^{k+1} - y^*(x^k)\|\|\bar{h}_f^k\|]$$
$$\overset{(f)}{\leq} 2\gamma_k\mathbb{E}[\|y^{k+1} - y^*(x^k)\|^2] + \frac{L_y^2\alpha_k^2}{8\gamma_k}\mathbb{E}[\|\bar{h}_f^k\|^2] \tag{49}$$

where (d) uses the fact that $\bar{h}_f^k = \mathbb{E}[h_f^k|\mathcal{F}_k']$; (e) follows from Lemma 4; and (f) uses the Young's inequality such that $ab \leq 2\gamma_k a^2 + \frac{b^2}{8\gamma_k}$.

Next we will use the smoothness of $y^*(x)$ in Lemma 2. We can bound $J_3^2$ by

$$- \mathbb{E}[\langle y^{k+1} - y^*(x^k), y^*(x^{k+1}) - y^*(x^k) - \nabla y^*(x^k)(x^{k+1} - x^k)\rangle]$$

$$\leq \mathbb{E}[\|y^{k+1} - y^*(x^k)\| \|y^*(x^{k+1}) - y^*(x^k) - \nabla y^*(x^k)(x^{k+1} - x^k)\|]$$

$$\overset{(g)}{\leq} \frac{L_{yx}}{2} \mathbb{E}\left[\|y^{k+1} - y^*(x^k)\| \|x^{k+1} - x^k\|^2\right]$$

$$\overset{(h)}{\leq} \frac{\eta L_{yx}}{4} \mathbb{E}\left[\|y^{k+1} - y^*(x^k)\|^2 \mathbb{E}[\|x^{k+1} - x^k\|^2|\mathcal{F}_k']\right] + \frac{L_{yx}}{4\eta} \mathbb{E}\left[\mathbb{E}[\|x^{k+1} - x^k\|^2|\mathcal{F}_k']\right]$$

$$\overset{(i)}{\leq} \frac{\eta L_{yx} \tilde{C}_f^2 \alpha_k^2}{4} \mathbb{E}[\|y^{k+1} - y^*(x^k)\|^2] + \frac{L_{yx}\alpha_k^2}{4\eta}(\mathbb{E}[\|\bar{h}_f^k\|^2] + \tilde{\sigma}_f^2) \tag{50}$$

where (g) uses the smoothness of $y^*(x)$; (h) follows from the Young's inequality such that $1 \leq \frac{\eta}{2} + \frac{1}{2\eta}$; and (i) uses the fact that $\mathbb{E}[\|h_f^k\|^2|\mathcal{F}_k'] \leq \tilde{C}_f^2$ in Lemma 2 and the variance bound in Lemma 5.

Plugging (49) and (50) into (48), we have

$$\mathbb{E}[J_3] \leq \left(2\gamma_k + \frac{\eta L_{yx}\tilde{C}_f^2}{4}\alpha_k^2\right)\mathbb{E}[\|y^{k+1} - y^*(x^k)\|^2] + \left(\frac{L_y^2\alpha_k^2}{8\gamma_k} + \frac{L_{yx}\alpha_k^2}{4\eta}\right)\mathbb{E}[\|\bar{h}_f^k\|^2] + \frac{L_{yx}\alpha_k^2}{4\eta}\tilde{\sigma}_f^2. \tag{51}$$

Plugging (47), (51) into (44), we get

$$\mathbb{E}[\|y^{k+1} - y^*(x^{k+1})\|^2] \leq \left(1 + 4\gamma_k + \frac{\eta L_{yx}\tilde{C}_f^2}{2}\alpha_k^2\right)\mathbb{E}[\|y^{k+1} - y^*(x^k)\|^2]$$

$$+ \left(L_y^2\alpha_k^2 + \frac{L_y^2\alpha_k^2}{4\gamma_k} + \frac{L_{yx}\alpha_k^2}{2\eta}\right)\mathbb{E}[\|\bar{h}_f^k\|^2] + \left(L_y^2\alpha_k^2 + \frac{L_{yx}\alpha_k^2}{2\eta}\right)\tilde{\sigma}_f^2$$

from which the proof is complete by choosing $\gamma_k = L_f L_y \alpha_k$.

## A.5 Proof of Theorem 1

Using Lemmas 1 and 3, we, respectively, bound the two difference terms in (15) and obtain

$$\mathbb{E}[\mathbb{V}^{k+1}] - \mathbb{E}[\mathbb{V}^k]$$

$$\leq - \frac{\alpha_k}{2}\mathbb{E}[\|\nabla F(x^k)\|^2] - \left(\frac{\alpha_k}{2} - \frac{L_F\alpha_k^2}{2} - \frac{L_f}{L_y}L_y^2\alpha_k^2 - \frac{L_f}{L_y}\frac{\alpha_k^2 L_y^2}{4\gamma_k} - \frac{L_f}{L_y}\frac{L_{yx}\alpha_k^2}{2\eta}\right)\mathbb{E}[\|\bar{h}_f^k\|^2]$$

$$+ \frac{L_f}{L_y}\left(1 + 4\gamma_k + L_f L_y\alpha_k + \frac{\eta L_{yx}\tilde{C}_f^2}{2}\alpha_k^2\right)\mathbb{E}[\|y^{k+1} - y^*(x^k)\|^2] - \frac{L_f}{L_y}\mathbb{E}[\|y^k - y^*(x^k)\|^2]$$

$$+ \alpha_k b_k^2 + \left(\frac{L_F}{2} + \frac{L_f}{L_y}L_y^2 + \frac{L_f}{L_y}\frac{L_{yx}}{2\eta}\right)\alpha_k^2\tilde{\sigma}_f^2$$

$$\overset{(a)}{\leq} - \frac{\alpha_k}{2}\mathbb{E}[\|\nabla F(x^k)\|^2] - \left(\frac{\alpha_k}{2} - \frac{L_F\alpha_k^2}{2} - L_f L_y\alpha_k^2 - \frac{L_f}{L_y}\frac{\alpha_k^2 L_y^2}{4\gamma_k} - \frac{L_f}{L_y}\frac{L_{yx}\alpha_k^2}{2\eta}\right)\mathbb{E}[\|\bar{h}_f^k\|^2]$$

$$+ \frac{L_f}{L_y}\left(\left(1 + 4\gamma_k + L_f L_y\alpha_k + \frac{\eta L_{yx}\tilde{C}_f^2}{2}\alpha_k^2\right)(1 - \rho_g\beta_k)^T - 1\right)\mathbb{E}[\|y^k - y^*(x^k)\|^2]$$

$$+ \frac{L_f}{L_y}\left(1 + 4\gamma_k + L_f L_y\alpha_k + \frac{\eta L_{yx}\tilde{C}_f^2}{2}\alpha_k^2\right)T\beta_k^2\sigma_{g,1}^2 + \alpha_k b_k^2 + \left(\frac{L_F}{2} + L_f L_y + \frac{L_f}{L_y}\frac{L_{yx}}{2\eta}\right)\alpha_k^2\tilde{\sigma}_f^2 \tag{52}$$

where (a) uses (18a) in Lemma 3.

Selecting $\gamma_k = L_f L_y \alpha_k$, we can simplify (52) as

$$\mathbb{E}[\mathbb{V}^{k+1}] - \mathbb{E}[\mathbb{V}^k] \leq - \frac{\alpha_k}{2}\mathbb{E}[\|\nabla F(x^k)\|^2] - \left(\frac{\alpha_k}{2} - \frac{L_F \alpha_k^2}{2} - L_f L_y \alpha_k^2 - \frac{\alpha_k}{4} - \frac{L_f}{L_y}\frac{L_{yx}\alpha_k^2}{2\eta}\right)\mathbb{E}[\|\bar{h}_f^k\|^2]$$

$$+ \frac{L_f}{L_y}\left(\left(1 + 2L_f L_y \alpha_k + \frac{\eta L_{yx}\tilde{C}_f^2}{2}\alpha_k^2\right)(1 - \rho_g \beta_k)^T - 1\right)\mathbb{E}[\|y^k - y^*(x^k)\|^2]$$

$$+ \frac{L_f}{L_y}\left(1 + 5L_f L_y \alpha_k + \frac{\eta L_{yx}\tilde{C}_f^2}{4}\alpha_k^2\right)T\beta_k^2\sigma_{g,1}^2$$

$$+ \alpha_k b_k^2 + \left(\frac{L_F}{2} + L_f L_y + \frac{L_{yx}L_f}{2\eta L_y}\right)\alpha_k^2\tilde{\sigma}_f^2. \tag{53}$$

To guarantee the descent of $\mathbb{V}^k$, the following constraints need to be satisfied

$$\alpha_k \leq \frac{1}{2L_F + 4L_f L_y + \frac{2L_f L_{yx}}{L_y \eta}} \tag{54a}$$

$$T\rho_g \beta_k \geq 2L_f L_y \alpha_k + \frac{\eta L_{yx}\tilde{C}_f^2}{2}\alpha_k^2 \tag{54b}$$

$$\beta_k \leq \frac{2}{\mu_g + \ell_{g,1}}. \tag{54c}$$

Finally, we define (with $\rho_g := \frac{2\mu_g \ell_{g,1}}{\mu_g + \ell_{g,1}}$)

$$\bar{\alpha}_1 = \frac{1}{2L_F + 4L_f L_y + \frac{2L_f L_{yx}}{L_y \eta}}, \quad \bar{\alpha}_2 = \frac{8T\rho_g}{(\mu_g + \ell_{g,1})(8L_f L_y + 2\eta L_{yx}\tilde{C}_f^2\bar{\alpha}_1)} \tag{55}$$

and, to satisfy the condition (54), we select the following stepsizes as

$$\alpha_k = \min\left\{\bar{\alpha}_1, \bar{\alpha}_2, \frac{\alpha}{\sqrt{K}}\right\}, \quad \beta_k = \frac{8L_f L_y + 2\eta L_{yx}\tilde{C}_f^2\bar{\alpha}_1}{4T\rho_g}\alpha_k. \tag{56}$$

With the above choice of stepsizes, (53) can be simplified as

$$\mathbb{E}[\mathbb{V}^{k+1}] - \mathbb{E}[\mathbb{V}^k] \leq - \frac{\alpha_k}{2}\mathbb{E}[\|\nabla F(x^k)\|^2] + c_1\alpha_k^2\sigma_{g,1}^2 + \alpha_k b_k^2 + c_2\alpha_k^2\tilde{\sigma}_f^2 \tag{57}$$

where the constants $c_1$ and $c_2$ are defined as

$$c_1 = \frac{L_f}{L_y}\left(1 + 5L_f L_y \bar{\alpha}_1 + \frac{\eta L_{yx}\tilde{C}_f^2}{4}\bar{\alpha}_1^2\right)\left(\frac{8L_f L_y + 2\eta L_{yx}\tilde{C}_f^2\bar{\alpha}_1}{4\rho_g}\right)^2\frac{1}{T}$$

$$c_2 = \left(\frac{L_F}{2} + L_f L_y + \frac{L_{yx}L_f}{2\eta L_y}\right). \tag{58}$$

Then telescoping leads to

$$\frac{1}{K}\sum_{k=0}^{K-1}\mathbb{E}[\|\nabla F(x^k)\|^2] \leq \frac{\mathbb{V}^0 + \sum_{k=0}^{K-1}\alpha_k b_k^2 + c_1\alpha_k^2\sigma_{g,1}^2 + c_2 T\beta_k^2\tilde{\sigma}_f^2}{\frac{1}{2}\sum_{k=0}^{K-1}\alpha_k}$$

$$\leq \frac{2\mathbb{V}^0}{K\min\{\bar{\alpha}_1, \bar{\alpha}_2\}} + \frac{2\mathbb{V}^0}{\alpha\sqrt{K}} + 2b_k^2 + \frac{2c_1\alpha}{\sqrt{K}}\sigma_{g,1}^2 + \frac{2c_2\alpha}{\sqrt{K}}\tilde{\sigma}_f^2. \tag{59}$$

To obtain the best $\kappa$-dependence, we choose the balancing constant $\eta = \frac{L_f}{L_y} = \mathcal{O}(\kappa)$, and then we can get $\bar{\alpha}_1 = \mathcal{O}(\kappa^{-3})$, $\bar{\alpha}_2 = \mathcal{O}(T\kappa^{-3})$, $c_1 = \mathcal{O}(\kappa^9/T)$, $c_2 = \mathcal{O}(\kappa^3)$. To obtain $b_k^2 = \frac{1}{\sqrt{K}}$, we need $N = \mathcal{O}(\kappa\log K)$. Select $\alpha = \Theta(\kappa^{-5/2})$ and $T = \mathcal{O}(\kappa^4)$, we are able to get

$$\frac{1}{K}\sum_{k=0}^{K-1}\mathbb{E}[\|\nabla F(x^k)\|^2] = \mathcal{O}\left(\frac{\kappa^3}{K} + \frac{\kappa^{5/2}}{\sqrt{K}}\right).$$

To achieve $\epsilon$-optimal solution, we need $K = \mathcal{O}(\kappa^5\epsilon^{-2})$, and the number of evaluations of $h_f^k, h_g^k$ are $\mathcal{O}(\kappa^5\epsilon^{-2}), \mathcal{O}(\kappa^9\epsilon^{-2})$ respectively.

# B Proof for stochastic min-max problem

Recall that the lower-level function for the min-max problem is $g(x, y; \phi) = -f(x, y; \xi)$. Then we rewrite the bilevel problem (1) as

$$\min_{x \in \mathbb{R}^d} \quad F(x) := \mathbb{E}_\xi \left[ f\left(x, y^*(x); \xi\right) \right] \tag{60a}$$

$$\text{s.t.} \quad y^*(x) = \arg\min_{y \in \mathbb{R}^{d'}} -\mathbb{E}_\xi[f(x, y; \xi)]. \tag{60b}$$

In this case, the bilevel gradient in (6) reduces to

$$\nabla F(x) := \nabla_x f\left(x, y^*(x)\right) + \nabla_x y^*(x)^\top \nabla_y f\left(x, y^*(x)\right) = \nabla_x f\left(x, y^*(x)\right) \tag{61}$$

where the second equality follows from the optimality condition of the lower-level problem, i.e., $\nabla_y f(x, y^*(x)) = 0$. We approximate $\nabla F(x)$ on a vector $y$ in place of $y^*(x)$, denoted as $\overline{\nabla} f(x, y) := \nabla_x f(x, y)$. Therefore, the alternating stochastic gradients for this special case are given by

$$h_g^{k,t} = -\nabla_y f(x^k, y^{k,t}; \xi_1^k) \quad \text{and} \quad h_f^k = \nabla_x f(x^k, y^{k+1}; \xi_2^k). \tag{62}$$

## B.1 Verifying lemmas

We make the following assumptions that are counterparts of Assumptions 1–3, most of which are common in the min-max optimization literature [9, 10, 30, 28].

**Assumption 4** (Lipschitz continuity). *Assume that $f(\cdot, y)$ is Lipchitz over $x \in \mathbb{R}^d$; that is, we have $\|f(x_1, y) - f(x_2, y)\| \le \ell_{f,0}\|x_1 - x_2\|$. Assume $\nabla f, \nabla^2 f$ are $\ell_{f,1}, \ell_{f,2}$-Lipschitz continuous; that is, for $z_1 := [x_1; y_1]$, $z_2 := [x_2; y_2]$, we have $\|\nabla f(x_1, y_1) - \nabla f(x_2, y_2)\| \le \ell_{f,1}\|z_1 - z_2\|, \|\nabla^2 f(x_1, y_1) - \nabla^2 f(x_2, y_2)\| \le \ell_{f,2}\|z_1 - z_2\|$.*

**Assumption 5** (Strong convexity of $f$ in $y$). *For any fixed $x$, $f(x, y)$ is $\mu_f$-strongly convex in $y$.*

Assumptions 1 and 2 together ensure that the first- and second-order derivations of $f(x, y)$ as well as the solution mapping $y^*(x)$ are well-behaved. Define the condition number $\kappa := \ell_{f,1}/\mu_f$.

**Assumption 6** (Stochastic derivatives). *The stochastic gradient $\nabla f(x, y; \xi)$ is an unbiased estimators of $\nabla f(x, y)$; and its variances is bounded by $\sigma_f^2$.*

Next we re-derive Lemmas 2, 4 and 5 for this special case.

**Lemma 6** (Counterparts of Lemmas 2, 4 and 5). *Under Assumptions 1–3, we have*

*(Lemma 2)* $\|\nabla y^*(x_1) - \nabla y^*(x_2)\| \le L_{yx}\|x_1 - x_2\|, \quad \mathbb{E}[\|h_f^k\|^2 | \mathcal{F}_k'] \le \tilde{C}_f^2$

*(Lemma 4)* $\|\overline{\nabla} f(x, y^*(x)) - \overline{\nabla} f(x, y)\| \le L_f \|y^*(x) - y\|$

$\quad\quad\quad\quad \|\nabla F(x_1) - \nabla F(x_2)\| \le L_F\|x_1 - x_2\|, \quad \|y^*(x_1) - y^*(x_2)\| \le L_y\|x_1 - x_2\|$

*(Lemma 5)* $\bar{h}_f^k = \overline{\nabla} f(x^k, y^{k+1}), \quad \mathbb{E}[\|h_f^k - \bar{h}_f^k\|^2 | \mathcal{F}_k'] \le \tilde{\sigma}_f^2$

*where the constants are defined as*

$$L_{yx} = \frac{\ell_{f,2} + \ell_{f,2}L_y}{\mu_f} + \frac{\ell_{f,1}(\ell_{f,2} + \ell_{f,2}L_y)}{\mu_f^2} = \mathcal{O}(\kappa^3), \quad \tilde{C}_f^2 = \ell_{l,0}^2 + \sigma_f^2$$

$$L_f = \ell_{f,1} = \mathcal{O}(1), \quad L_F = (\ell_{f,1} + \frac{\ell_{f,1}^2}{\mu_f}) = \mathcal{O}(\kappa), \quad L_y = \frac{\ell_{f,1}}{\mu_f} = \mathcal{O}(\kappa), \quad \tilde{\sigma}_f^2 = \sigma_f^2.$$

**Proof:** We first calculate $L_f$ by

$$\|\overline{\nabla} f(x, y^*(x)) - \overline{\nabla} f(x, y)\| = \|\nabla_x f(x, y^*(x)) - \nabla_x f(x, y)\|$$
$$\le \ell_{f,1}\|y^*(x) - y\| := L_f\|y^*(x) - y\|. \tag{63}$$

We then calculate $L_F$ by

$$\|\nabla F(x_1) - \nabla F(x_2)\| = \|\nabla_x f(x_1, y^*(x_1)) - \nabla_x f(x_2, y^*(x_2))\|$$
$$\le \|\nabla_x f(x_1, y^*(x_1)) - \nabla_x f(x_2, y^*(x_1))\| + \|\nabla_x f(x_2, y^*(x_1)) - \nabla_x f(x_2, y^*(x_2))\|$$
$$\le \ell_{f,1}\|x_1 - x_2\| + \ell_{f,1}\|y^*(x_1) - y^*(x_2)\|$$
$$\le \left(\ell_{f,1} + \frac{\ell_{f,1}^2}{\mu_f}\right)\|x_1 - x_2\| := L_F\|x_1 - x_2\|. \tag{64}$$

The calculation of $L_y, L_{yx}$ follows the proof of Lemma 2 and Lemma 4, and $\tilde{\sigma}_f^2, \tilde{C}_f^2, \sigma_g^2$ follows from the fact $h_f^k = \nabla_x f(x^k, y^{k+1}; \xi_2^k), h_g^{k,t} = -\nabla_y f(x^k, y^{k,t}; \xi_2^{k,t})$. Note that different from the bilevel case, the upper-level gradients $h_f^k$ in the min-max case only contain $\nabla_x f$ not $\nabla_y f$, which only needs the lipschitz continuity of $x$.

### B.2 Reduction from Theorem 1 to Proposition 3

In the min-max case, we apply Theorem 1 with $\eta = 1$. We define

$$\bar{\alpha}_1 = \frac{1}{2L_F + 4L_f L_y + \frac{L_f L_{yx}}{L_y}}, \quad \bar{\alpha}_2 = \frac{8T\rho_g}{(\mu_g + \ell_{g,1})(8L_f L_y + L_{yx}\tilde{C}_f^2\bar{\alpha}_1)}$$

and, to satisfy the condition (54), we select

$$\alpha_k = \min\{\bar{\alpha}_1, \bar{\alpha}_2, \frac{\alpha}{\sqrt{K}}\} \quad \text{and} \quad \beta_k = \frac{8L_f L_y + L_{yx}\tilde{C}_f^2\bar{\alpha}_1}{4T\rho_g}\alpha_k.$$

With the above choice of stepsizes, (59) can be simplified as

$$\frac{1}{K}\sum_{k=0}^{K-1}\mathbb{E}[\|\nabla F(x^k)\|^2] \leq \frac{2\mathbb{V}^0}{K\min\{\bar{\alpha}_1, \bar{\alpha}_2\}} + \frac{2\mathbb{V}^0}{\alpha\sqrt{K}} + \frac{2c_1\alpha}{\sqrt{K}}\sigma_g^2 + \frac{2c_2\alpha}{\sqrt{K}}\sigma_f^2, \tag{65}$$

where the constants can be defined as

$$c_1 = \frac{L_f}{L_y}\left(1 + 2L_f L_y\alpha_k + \frac{L_{yx}\tilde{C}_f^2}{4}\alpha_k^2\right)\left(\frac{8L_f L_y + \eta L_{yx}\tilde{C}_f^2\bar{\alpha}_1}{4\rho_g}\right)^2\frac{1}{T} = \mathcal{O}(\frac{\kappa^3}{T})$$

$$c_2 = \left(\frac{L_F}{2} + L_f L_y + \frac{L_{yx}L_f}{4L_y}\right) = \mathcal{O}(\kappa^2).$$

Note that $\bar{\alpha}_1 = \mathcal{O}(\kappa^{-2}), \bar{\alpha}_2 = \mathcal{O}(T\kappa^{-2})$. Select $\alpha = \Theta(\kappa^{-1}), T = \Theta(\kappa)$, then

$$\frac{1}{K}\sum_{k=0}^{K-1}\mathbb{E}[\|\nabla F(x^k)\|^2] = \mathcal{O}\left(\frac{\kappa^2}{K} + \frac{\kappa}{\sqrt{K}}\right). \tag{66}$$

To achieve $\epsilon$-accuracy, we need $K = \mathcal{O}(\kappa^2\epsilon^{-2})$. And the number of gradient evaluations for $h_f^k, h_g^k$ are $\mathcal{O}(\kappa^2\epsilon^{-2}), \mathcal{O}(\kappa^3\epsilon^{-2})$ respectively.

## C   Proof for stochastic compositional problem

Recall that in the stochastic compositional problem, the upper-level function is defined as $f(x, y; \xi) := f(y; \xi)$, and the lower-level function is defined as $g(x, y; \phi) := \frac{1}{2}\|y - h(x; \phi)\|^2$. Then we rewrite the bilevel problem (1) as

$$\min_{x \in \mathbb{R}^d} \quad F(x) := \mathbb{E}_\xi\left[f\left(y^*(x); \xi\right)\right] \tag{67a}$$

$$\text{s.t.} \quad y^*(x) = \arg\min_{y \in \mathbb{R}^{d'}} \frac{1}{2}\mathbb{E}_\phi[\|y - h(x; \phi)\|^2]. \tag{67b}$$

In this case, the bilevel gradient in (6) reduces to

$$\nabla F(x) := \nabla_{xy}^2 g(x, y^*(x))\left[\nabla_{yy}^2 g(x, y^*(x))\right]^{-1}\nabla_y f(x, y^*(x))$$

$$= \nabla h(x; \phi)^\top \nabla_y f(y^*(x)) \tag{68}$$

where we use the fact that $\nabla_{yy}^2 g(x, y; \phi) = \mathbf{I}_{d' \times d'}$ and $\nabla_{xy}^2 g(x, y; \phi) = -\nabla h(x; \phi)^\top$.

Similar to Section 2, we again evaluate $\nabla F(x)$ on a certain vector $y$ in place of $y^*(x)$, which is denoted as $\overline{\nabla} f(x, y) = \nabla h(x)\nabla f(y)$. Therefore, the alternating stochastic gradients $h_f^k, h_g^{k,t}$ for this special case are much simpler, given by

$$h_g^{k,t} = y^{k,t} - h(x^k; \phi^{k,t}) \quad \text{and} \quad h_f^k = \nabla h(x^k; \phi^k)\nabla f(y^{k+1}; \xi^k). \tag{69}$$

It can be observed that $h_f^k$ is an unbiased estimate of $\overline{\nabla} f(x^k, y^{k+1})$, that is, $\bar{h}_f^k = \overline{\nabla} f(x, y), b_k = 0$.

## C.1 Verifying lemmas

We make the following assumptions that are counterparts of Assumptions 1–3, all of which are common in compositional optimization literature [12, 37, 14, 41, 38].

**Assumption 7** (Lipschitz continuity). *Assume that $f, \nabla f, h, \nabla h$ are respectively $\ell_{f,0}, \ell_{f,1}, \ell_{h,0}, \ell_{h,1}$-Lipschitz continuous; that is, for $z_1 := [x_1; y_1]$, $z_2 := [x_2; y_2]$, we have $\|f(x_1, y_1) - f(x_2, y_2)\| \leq \ell_{f,0}\|z_1 - z_2\|$, $\|\nabla f(x_1, y_1) - \nabla f(x_2, y_2)\| \leq \ell_{f,1}\|z_1 - z_2\|$, $\|h(x_1) - h(x_2)\| \leq \ell_{h,0}\|x_1 - x_2\|$, $\|\nabla h(x_1) - \nabla h(x_2)\| \leq \ell_{h,1}\|x_1 - x_2\|$.*

Note that the Lipschitz continuity of $\nabla g, \nabla^2 g$ in Assumption 1 can be implied by the Lipschitz continuity of $h, \nabla h$ in the above assumption. Assumption 2 is automatically satisfied for stochastic compositional problems since $\nabla_{yy} g(x, y; \phi) = \mathbf{I}_{d' \times d'}$ and the condition number $\kappa := 1$.

**Assumption 8** (Stochastic derivatives). *The stochastic quantities $\nabla f(x, y; \xi)$, $h(x; \phi)$, $\nabla h(x; \phi)$ are unbiased estimators of $\nabla f(x, y)$, $h(x)$, $\nabla h(x)$, respectively; and their variances are bounded by $\sigma_f^2, \sigma_{h,0}^2, \sigma_{h,1}^2$, respectively.*

The unbiasedness and bounded variance of $\nabla g(x, y; \phi), \nabla^2 g(x, y, \phi)$ in Assumption 3 can be implied by the unbiasedness and bounded variance of $h(x; \phi), \nabla h(x; \phi)$.

Next we re-derive Lemmas 2, 4 and 5 for this special case.

**Lemma 7** (Counterparts of Lemmas 2, 4 and 5). *Under Assumptions 1–3, we have*

*(Lemma 2)* $\|\nabla y^*(x_1) - \nabla y^*(x_2)\| \leq L_{yx}\|x_1 - x_2\|, \quad \mathbb{E}[\|h_f^k\|^2 | \mathcal{F}_k'] \leq \tilde{C}_f^2$

*(Lemma 4)* $\|\overline{\nabla} f(x, y^*(x)) - \overline{\nabla} f(x, y)\| \leq L_f\|y^*(x) - y\|$
$\qquad\quad \|\nabla F(x_1) - \nabla F(x_2)\| \leq L_F\|x_1 - x_2\|, \quad \|y^*(x_1) - y^*(x_2)\| \leq L_y\|x_1 - x_2\|$

*(Lemma 5)* $\mathbb{E}[\|h_f^k - \bar{h}_f^k\|^2 | \mathcal{F}_k'] \leq \tilde{\sigma}_f^2, \quad \bar{h}_f^k = \overline{\nabla} f(x^k, y^{k+1})$

*where the constants are defined as*

$$L_f = \ell_{h,0}\ell_{f,1}, \quad L_y = \ell_{h,0}, \quad L_F = \ell_{h,0}^2\ell_{f,1} + \ell_{f,0}\ell_{h,1}, \quad L_{yx} = \ell_{h,1}$$
$$\tilde{\sigma}_f^2 = \ell_{h,0}^2\sigma_f^2 + (\ell_{f,0}^2 + \sigma_f^2)\sigma_{h,1}^2, \quad \tilde{C}_f^2 = (\ell_{f,0}^2 + \sigma_f^2)(\ell_{h,0}^2 + \sigma_{h,1}^2). \tag{70}$$

**Proof:** We first calculate $L_f$ by

$$\begin{aligned}
\|\overline{\nabla} f(x, y^*(x)) - \overline{\nabla} f(x, y)\| &= \|\nabla h(x)\nabla f(y^*(x)) - \nabla h(x)\nabla f(y)\| \\
&\leq \|\nabla h(x)\|\|\nabla f(y^*(x)) - \nabla f(y)\| \\
&\leq \ell_{h,0}\ell_{f,1}\|y^*(x) - y\| := L_f\|y^*(x) - y\|.
\end{aligned}$$

We then calculate $L_F$ by

$$\begin{aligned}
\|\nabla F(x_1) - \nabla F(x_2)\| &= \|\nabla h(x_1)\nabla f(h(x_1)) - \nabla h(x_2)\nabla f(h(x_2))\| \\
&\leq \|\nabla h(x_1)\|\|\nabla f(h(x_1)) - \nabla f(h(x_2))\| + \|\nabla f(h(x_2))\|\|\nabla h(x_1) - \nabla h(x_2)\| \\
&\leq \ell_{h,0}^2\ell_{f,1}\|x_1 - x_2\| + \ell_{f,0}\ell_{h,1}\|x_1 - x_2\| \\
&:= L_F\|x_1 - x_2\|. \tag{71}
\end{aligned}$$

We then calculate $L_y$ and $L_{yx}$ by

$$\begin{aligned}
\|y^*(x_1) - y^*(x_2)\| &= \|h(x_1) - h(x_2)\| \leq \ell_{h,0}\|x_1 - x_2\| := L_y\|x_1 - x_2\| \\
\|\nabla y^*(x_1) - \nabla y^*(x_2)\| &= \|\nabla h(x_1) - \nabla h(x_2)\| \leq \ell_{h,1}\|x_1 - x_2\| := L_{yx}\|x_1 - x_2\|.
\end{aligned}$$

We then calculate $\tilde{\sigma}_f^2$ by

$$\begin{aligned}
\mathbb{E}[\|h_f^k - \bar{h}_f^k\|^2 | \mathcal{F}_k'] &\leq \mathbb{E}[\|\nabla h(x^k; \phi_2^k)\nabla f(y^{k+1}; \xi^k) - \nabla h(x^k)\nabla f(y^{k+1})\|^2 | \mathcal{F}_k'] \\
&\leq \mathbb{E}[\|\nabla f(y^{k+1}; \xi^k)\|^2\|\nabla h(x^k; \phi_2^k) - \nabla h(x^k)\|^2 | \mathcal{F}_k'] \\
&\quad + \mathbb{E}[\|\nabla h(x^k)\|^2\|\nabla f(y^{k+1}; \xi^k) - \nabla f(y^{k+1})\|^2 | \mathcal{F}_k'] \\
&\leq (\ell_{f,0}^2 + \sigma_f^2)\sigma_{h,1}^2 + \ell_{h,0}^2\sigma_f^2 := \tilde{\sigma}_f^2. \tag{72}
\end{aligned}$$

We then calculate $\tilde{C}_f^2$ by

$$
\begin{aligned}
\mathbb{E}[\|h_f^k\|^2|\mathcal{F}_k'] &= \mathbb{E}[\|\nabla h(x^k;\phi_2^k)\nabla f(y^{k+1};\xi^k)\|^2|\mathcal{F}_k'] \\
&\leq \mathbb{E}[\|\nabla f(y^{k+1};\xi^k)\|^2|\mathcal{F}_k']\mathbb{E}[\|\nabla h(x^k;\phi_2^k)\|^2|\mathcal{F}_k'] \\
&\leq (\ell_{f,0}^2 + \sigma_f^2)(\ell_{h,0}^2 + \sigma_{h,1}^2) := \tilde{C}_f^2.
\end{aligned}
\tag{73}
$$

### C.2 Reduction from Theorem 1 to Proposition 4

In the compositional case, we apply Theorem 1 by setting $T = 1, \alpha = 1, \eta = \frac{1}{\ell_{h,1}}$. We define

$$
\bar{\alpha}_1 = \frac{1}{6\ell_{h,0}^2\ell_{f,1} + 2\ell_{f,0}\ell_{h,1} + \ell_{f,1}\ell_{h,1}^2}, \quad \bar{\alpha}_2 = \frac{8}{(\mu_g + \ell_{g,1})(8\ell_{f,1}\ell_{h,0}^2 + \tilde{C}_f^2\bar{\alpha}_1)}
$$

and, to satisfy the condition (54), we select

$$
\alpha_k = \min\left\{\bar{\alpha}_1, \bar{\alpha}_2, \frac{\alpha}{\sqrt{K}}\right\} \quad \text{and} \quad \beta_k = \frac{8\ell_{f,1}\ell_{h,0}^2 + \tilde{C}_f^2\bar{\alpha}_1}{4}\alpha_k.
\tag{74}
$$

And the constants $c_1, c_2$ in (58) reduce to

$$
\begin{aligned}
c_1 &= \ell_{f,1}\left(1 + 2\ell_{f,1}\ell_{h,0}^2\bar{\alpha}_1 + \frac{\tilde{C}_f^2}{4}\bar{\alpha}_1^2\right)\left(\frac{8\ell_{h,0}^2\ell_{f,1} + \tilde{C}_f^2\bar{\alpha}_1}{4}\right)^2 \\
c_2 &= \left(\frac{\ell_{h,0}^2\ell_{f,1} + \ell_{f,0}\ell_{h,1}}{2} + \ell_{h,0}^2\ell_{f,1} + \frac{\ell_{f,1}\ell_{h,1}^2}{4}\right).
\end{aligned}
\tag{75}
$$

We apply (59) and get

$$
\frac{1}{K}\sum_{k=0}^{K-1}\mathbb{E}[\|\nabla F(x^k)\|^2] \leq \frac{2\mathbb{V}^0}{K\min\{\bar{\alpha}_1, \bar{\alpha}_2\}} + \frac{2\mathbb{V}^0}{\alpha\sqrt{K}} + \frac{2c_1}{\sqrt{K}}\sigma_{h,1}^2 + \frac{2c_2}{\sqrt{K}}\tilde{\sigma}_f^2 = \mathcal{O}\left(\frac{1}{\sqrt{K}}\right)
\tag{76}
$$

from which the proof is complete.

## D  Proof for actor-critic method

Recall the state feature mapping $\phi(\cdot): \mathcal{S} \to \mathbb{R}^{d'}$. Define

$$
A_{\theta,\phi} := \mathbb{E}_{s \sim \mu_\theta, s' \sim \mathcal{P}_{\pi_\theta}}[\phi(s)(\gamma\phi(s') - \phi(s))^\top],
\tag{77a}
$$

$$
b_{\theta,\phi} := \mathbb{E}_{s \sim \mu_\theta, a \sim \pi_\theta, s' \sim \mathcal{P}}[r(s,a,s')\phi(s)].
\tag{77b}
$$

It is known that for a given $\theta$, the stationary point $y^*(\theta)$ of the TD update in (29) satisfies

$$
A_{\theta,\phi}y^*(\theta) + b_{\theta,\phi} = 0.
\tag{78}
$$

Due to the special nature of the policy gradient, we make the following assumptions that will lead to the counterparts of Lemmas 2, 4 and 5 in reinforcement learning. These assumptions are mostly common in analyzing actor-critic method with linear value function approximation [50–52].

**Assumption 9.** *For all $s \in \mathcal{S}$, the feature vector $\phi(s)$ is normalized so that $\|\phi(s)\|_2 \leq 1$. For all eligible $\theta$, $A_{\theta,\phi}$ is negative definite and its maximum eigenvalue is upper bounded by constant $-\lambda$.*

Assumption 9 is common in analyzing TD with linear function approximation; see e.g., [54, 55, 50]. With this assumption, $A_{\theta,\phi}$ is invertible, so we have $y^*(\theta) = -A_{\theta,\phi}^{-1}b_{\theta,\phi}$. Defining $R_y := r_{\max}/\lambda$, we have $\|y^*(\theta)\|_2 \leq R_y$. It justifies the projection introduced in the critic update (29).

**Assumption 10.** *For any $\theta, \theta' \in \mathbb{R}^d$, $s \in \mathcal{S}$ and $a \in \mathcal{A}$, there exist constants $C_\psi, L_\psi, L_\pi$ such that: i) $\|\psi_\theta(s,a)\|_2 \leq C_\psi$; ii) $\|\psi_\theta(s,a) - \psi_{\theta'}(s,a)\|_2 \leq L_\psi\|\theta - \theta'\|_2$; iii) $|\pi_\theta(a|s) - \pi_{\theta'}(a|s)| \leq L_\pi\|\theta - \theta'\|_2$.*

Assumption 10 is common in analyzing policy gradient-type algorithms which has also been made by e.g., [56, 57]. This assumption holds for many policy parameterization methods such as tabular softmax policy [57], Gaussian policy [58] and Boltzmann policy [44].

**Assumption 11.** *For any $\theta, \theta' \in \mathbb{R}^d$, there exist constants such that: i) $\|\nabla \mu_\theta(s)\|_2 \leq C_\mu$; ii) $\|\nabla \mu_\theta(s) - \nabla \mu_{\theta'}(s)\|_2 \leq L_{\mu,1}\|\theta - \theta'\|_2$; iii) $|\mu_\theta(s) - \mu_{\theta'}(s)| \leq L_{\mu,0}\|\theta - \theta'\|_2$.*

Assumption 11 is the counterpart of Assumption 10 that is made for the stationary distribution $\mu_\theta(a|s)$. Note that the existence of $\nabla \mu_\theta(s)$ has been shown in [59]. In this case, under Assumption 10, i) and iii) of Assumption 11 can be obtained from the sensitivity analysis of Markov chain; see e.g., [60, Theorem 3.1]. While we cannot provide a justification of (ii), we found it necessary to ensure the smoothness of the lower-level critic solution $y^*(\theta)$.

**Assumption 12.** *For any $\theta$, the Markov chain under $\pi_\theta$ and transition kernel $\mathcal{P}(\cdot|s, a)$ is irreducible and aperiodic. Then there exist constants $\kappa > 0$ and $\rho \in (0, 1)$ such that*

$$\sup_{s \in \mathcal{S}} d_{TV}\left(\mathbb{P}(s_t \in \cdot | s_0 = s, \pi_\theta), \mu_\theta\right) \leq \kappa \rho^t, \quad \forall t \tag{79}$$

*where $\mu_\theta$ is the stationary state distribution under $\pi_\theta$, and $s_t$ is the state of Markov chain at time $t$.*

Assumption 12 assumes the Markov chain mixes at a geometric rate; see also [54, 55].

We define the critic approximation error as

$$\epsilon_{app} := \max_{\theta \in \mathbb{R}^d} \sqrt{\mathbb{E}_{s \sim \mu_\theta} |V_{\pi_\theta}(s) - \hat{V}_{y^*_\theta}(s)|^2}. \tag{80}$$

This error captures the quality of the critic function approximation; see also [61, 50, 51]. It becomes zero when the value function $V_{\pi_\theta}$ belongs to the linear function space for any $\theta$.

### D.1 Auxiliary lemmas

We give a proposition regarding the $L_F$-Lipschitz of the policy gradient under proper assumptions.

**Proposition 6** (Smoothness of policy gradiemt [56]). *Suppose Assumption 10 holds. For any $\theta, \theta' \in \mathbb{R}^d$, we have $\|\nabla F(\theta) - \nabla F(\theta')\|_2 \leq L_F \|\theta - \theta'\|_2$, where $L_F$ is a positive constant.*

We provide a justification for Lipschitz continuity of $y^*(\theta)$ in the next proposition.

**Proposition 7** (Lipschitz continuity of $y^*(\theta)$). *Suppose Assumption 10 and 12 hold. For any $\theta_1, \theta_2 \in \mathbb{R}^d$, we have $\|y^*(\theta_1) - y^*(\theta_2)\|_2 \leq L_y \|\theta_1 - \theta_2\|_2$, where $L_y$ is a positive constant.*

*Proof.* We use $y_1^*, y_2^*, A_1, A_2, b_1$ and $b_2$ as shorthand notations of $y^*(\theta_1)$, $y^*(\theta_2)$, $A_{\pi_{\theta_1}}$, $A_{\pi_{\theta_2}}$, $b_{\pi_{\theta_1}}$ and $b_{\pi_{\theta_2}}$ respectively. By Assumption 9, $A_{\theta,\phi}$ is invertible for any $\theta \in \mathbb{R}^d$, so we can write $y^*(\theta) = -A_{\theta,\phi}^{-1} b_{\theta,\phi}$. Then we have

$$
\begin{aligned}
\|y_1^* - y_2^*\|_2 &= \| - A_1^{-1}b_1 + A_2^{-1}b_2\|_2 \\
&= \| - A_1^{-1}b_1 - A_1^{-1}b_2 + A_1^{-1}b_2 + A_2^{-1}b_2\|_2 \\
&= \| - A_1^{-1}(b_1 - b_2) - (A_1^{-1} - A_2^{-1})b_2\|_2 \\
&\leq \|A_1^{-1}(b_1 - b_2)\|_2 + \|(A_1^{-1} - A_2^{-1})b_2\|_2 \\
&\leq \|A_1^{-1}\|_2\|b_1 - b_2\|_2 + \|A_1^{-1} - A_2^{-1}\|_2\|b_2\|_2 \\
&= \|A_1^{-1}\|_2\|b_1 - b_2\|_2 + \|A_1^{-1}(A_2 - A_1)A_2^{-1}\|_2\|b_2\|_2 \\
&\leq \|A_1^{-1}\|_2\|b_1 - b_2\|_2 + \|A_1^{-1}\|_2\|A_2^{-1}\|_2\|b_2\|_2\|(A_2 - A_1)\|_2 \\
&\leq \lambda^{-1}\|b_1 - b_2\|_2 + \lambda^{-2}r_{\max}\|A_1 - A_2\|_2,
\end{aligned}
\tag{81}
$$

where the last inequality follows Assumption 9, and the fact that

$$\|b_2\|_2 = \|\mathbb{E}[r(s, a, s')\phi(s)]\|_2 \leq \mathbb{E}\|r(s, a, s')\phi(s)\|_2 \leq \mathbb{E}[|r(s, a, s')|\|\phi(s)\|_2] \leq r_{\max}. \tag{82}$$

Denote $(s^1, a^1, s'^1)$ and $(s^2, a^2, s'^2)$ as samples drawn with $\theta_1$ and $\theta_2$ respectively, i.e. $s^1 \sim \mu_{\theta_1}$, $a^1 \sim \pi_{\theta_1}$, $s'^1 \sim \mathcal{P}$ and $s^2 \sim \mu_{\theta_2}$, $a^2 \sim \pi_{\theta_2}$, $s'^2 \sim \mathcal{P}$. Then we have

$$
\begin{aligned}
\|b_1 - b_2\|_2 &= \left\|\mathbb{E}\left[r(s^1, a^1, s'^1)\phi(s^1)\right] - \mathbb{E}\left[r(s^2, a^2, s'^2)\phi(s^2)\right]\right\|_2 \\
&\leq \sup_{s,a,s'} \|r(s, a, s')\phi(s)\|_2 \|\mathbb{P}((s^1, a^1, s'^1) \in \cdot) - \mathbb{P}((s^2, a^2, s'^2) \in \cdot)\|_{TV} \\
&\leq r_{\max}\|\mathbb{P}((s^1, a^1, s'^1) \in \cdot) - \mathbb{P}((s^2, a^2, s'^2) \in \cdot)\|_{TV} \\
&= 2r_{\max}d_{TV}\left(\mu_{\theta_1} \otimes \pi_{\theta_1} \otimes \mathcal{P}, \mu_{\theta_2} \otimes \pi_{\theta_2} \otimes \mathcal{P}\right) \\
&\leq 2r_{\max}|\mathcal{A}|L_\pi(1 + \log_\rho \kappa^{-1} + (1 - \rho)^{-1})\|\theta_1 - \theta_2\|_2,
\end{aligned}
\tag{83}
$$

where the first inequality follows the definition of total variation (TV) norm, and the last inequality follows in [50, Lemma A.1]. Similarly we have:

$$
\begin{aligned}
\|A_1 - A_2\|_2 &\leq 2(1+\gamma)d_{TV}\left(\mu_{\theta_1} \otimes \pi_{\theta_1}, \mu_{\theta_2} \otimes \pi_{\theta_2}\right) \\
&= (1+\gamma)|\mathcal{A}|L_\pi(1 + \log_\rho \kappa^{-1} + (1-\rho)^{-1})\|\theta_1 - \theta_2\|_2 \\
&:= L_{A,0}\|\theta_1 - \theta_2\|_2.
\end{aligned}
\tag{84}
$$

Substituting (83) and (84) into (81) completes the proof. $\qquad\square$

We prove the Lipschitz continuity of $\nabla_\theta y^*(\theta)$ next, for which we will use the following fact.

**Fact.** If the functions $f(\theta), g(\theta)$ are bounded by $C_f$ and $C_g$; and are $L_f$- and $L_g$-Lipschitz continuous, then $f(\theta)g(\theta)$ is also bounded by $C_f C_g$ and is $(C_f L_g + C_g L_f)$-Lipschitz continuous.

*Proof.* Using the Cauchy-Schwartz inequality, it is easy to see that $f(\theta), g(\theta)$ are bounded by $C_f C_g$. In addition, we have that

$$
\begin{aligned}
\|f(\theta_1)g(\theta_1) - f(\theta_2)g(\theta_2)\| &= \|f(\theta_1)g(\theta_1) - f(\theta_1)g(\theta_2) + f(\theta_1)g(\theta_2) - f(\theta_2)g(\theta_2)\| \\
&\leq \|f(\theta_1)\|\|g(\theta_1) - g(\theta_2)\| + \|f(\theta_1) - f(\theta_2)\|\|g(\theta_2)\| \\
&\leq (C_f L_g + C_g L_f)\|\theta_1 - \theta_2\|_2
\end{aligned}
$$

which implies that $f(\theta), g(\theta)$ is $(C_f L_g + C_g L_f)$-Lipschitz continuous.

**Proposition 8** (Lipschitz continuity of $\nabla_\theta y^*(\theta)$). *Suppose Assumption 10-12 hold. For any $\theta_1, \theta_2 \in \mathbb{R}^d$, we have $\|\nabla_\theta y^*(\theta_1) - \nabla_\theta y^*(\theta_2)\|_2 \leq L_{yx}\|\theta_1 - \theta_2\|_2$, where $L_{yx}$ is a positive constant.*

*Proof.* With $y^*(\theta) = -A_{\theta,\phi}^{-1}b_{\theta,\phi}$, we have

$$
\nabla_\theta y^*(\theta) = -\nabla_\theta(A_{\theta,\phi}^{-1}b_{\theta,\phi}) = -A_{\theta,\phi}^{-1}(\nabla_\theta A_{\theta,\phi})A_{\theta,\phi}^{-1}b_{\theta,\phi} - A_{\theta,\phi}(\nabla_\theta b_{\theta,\phi}).
\tag{85}
$$

To validate the Lipschitz continuity of $\nabla_\theta y^*(\theta)$, we need to show the boundedness and Lipschitz continuity of $A_{\theta,\phi}^{-1}$, $b_{\theta,\phi}$, $\nabla_\theta A_{\theta,\phi}$ and $\nabla_\theta b_{\theta,\phi}$.

From (83) and (84), we have that there exist constants $L_{A,0}$ and $L_{b,0}$ such that $A_{\theta,\phi}$ is $L_{A,0}$-Lipschitz continuous, and $b_{\theta,\phi}$ is $L_{b,0}$-Lipschitz continuous. From Assumption 9 and (82), we have that there exist constants $C_{A,0}$ and $C_{b,0}$ such that $\|A_{\theta,\phi}\|_2 \leq C_{A,0}$, and $\|b_{\theta,\phi}\|_2 \leq C_{b,0}$.

In addition, using $A_1$ and $A_2$ as shorthand notations of $A_{\pi_{\theta_1}}$ and $A_{\pi_{\theta_2}}$, respectively, we have

$$
\begin{aligned}
\|A_1^{-1} - A_2^{-1}\|_2 &= \|A_1^{-1}(A_2 - A_1)A_2^{-1}\|_2 \\
&\leq \|A_1^{-1}\|_2\|A_2^{-1}\|_2\|(A_2 - A_1)\|_2 \\
&\leq \lambda^{-2}\|A_1 - A_2\|_2 \\
&\overset{(84)}{\leq} \lambda^{-2}L_{A,0}\|\theta_1 - \theta_2\|_2.
\end{aligned}
\tag{86}
$$

Therefore, $A_{\theta,\phi}^{-1}$ is $\lambda^{-2}L_{A,0}$-Lipschitz continuous, and is bounded by $\lambda^{-1}$ due to Assumption 9.

For simplicity, denote

$$
A(s, s') := \phi(s)(\gamma\phi(s') - \phi(s))^\top, \quad b(s, a, s') := r(s, a, s')\phi(s)
\tag{87}
$$

and then $b_{\theta,\phi} := \mathbb{E}_{s\sim\mu_\theta, a\sim\pi_\theta, s'\sim\mathcal{P}}[b(s, a, s')]$ and $A_{\theta,\phi} := \mathbb{E}_{s\sim\mu_\theta, s'\sim\mathcal{P}_{\pi_\theta}}[A(s, s')]$.

Next we analyze $\nabla_\theta A_{\theta,\phi}$ and $\nabla_\theta b_{\theta,\phi}$, which is given by

$$
\begin{aligned}
\nabla_\theta A_{\theta,\phi} &= \nabla_\theta\Big(\sum_{s,a,s'} \mu_\theta(s)\pi_\theta(a|s)P(s'|s,a)A(s,s')\Big) \\
&= \sum_{s,a,s'}[\nabla_\theta\mu_\theta(s)\pi_\theta(a|s)P(s'|s,a)A(s,s') + \mu_\theta(s)\nabla_\theta\pi_\theta(a|s)P(s'|s,a)A(s,s')].
\end{aligned}
\tag{88}
$$

From Assumption 10 and 11, $\mu_\theta(s), \pi_\theta(a|s), \nabla_\theta\mu_\theta(s), \nabla_\theta\pi_\theta(a|s)$ are Lipschitz continuous and bounded. Using the **Fact**, we can show that there exist constants $C_{A,1}$ and $L_{A,1}$ such that $\nabla_\theta A_{\theta,\phi}$ is $L_{A,1}$-Lipschitz continuous and bounded by $C_{A,1}$.

Likewise, we have

$$\nabla_\theta b_{\theta,\phi} = \nabla_\theta \Big( \sum_{s,a,s'} \mu_\theta(s)\pi_\theta(a|s)P(s'|s,a)b(s,a,s') \Big) \tag{89}$$

$$= \sum_{s,a,s'} \left[ \nabla_\theta\mu_\theta(s)\pi_\theta(a|s)P(s'|s,a)b(s,a,s') + \mu_\theta(s)\nabla_\theta\pi_\theta(a|s)P(s'|s,a)b(s,a,s') \right].$$

From Assumption 10 and 11, $\mu_\theta(s), \pi_\theta(a|s), \nabla_\theta\mu_\theta(s), \nabla_\theta\pi_\theta(a|s)$ are Lipschitz continuous and bounded. Using the **Fact**, we are able to show that there exist constants $C_{b,1}$ and $L_{b,1}$ such that $\nabla_\theta b_{\theta,\phi}$ is $L_{b,1}$-Lipschitz continuous and bounded by $C_{b,1}$.

Therefore, since $A_{\theta,\phi}^{-1}$, $b_{\theta,\phi}$, $\nabla_\theta A_{\theta,\phi}$ and $\nabla_\theta b_{\theta,\phi}$ are all Lipschitz continuous, using **Fact**, we can show that $\nabla_\theta y^*(\theta)$ in (85) is $L_{yx}$-Lipschitz continuous, where $L_{yx}$ depends on the constants $C_\mu, C_\psi, L_\pi, L_{\mu,0}, L_{\mu,1}, \lambda$ defined in Assumptions 10-12. $\qquad\square$

### D.2 Convergence of critic variables

For brevity, we first define the following notations (cf. $\xi := (s,a,s')$):

$$\hat{\delta}(\xi,y) := r(s,a,s') + \gamma\phi(s')^\top y - \phi(s)^\top y,$$
$$h_g(\xi,y) := \hat{\delta}(\xi,y)\phi(s),$$
$$\overline{h}_g(\theta,y) := \mathbb{E}_{s\sim\mu_\theta,a\sim\pi_\theta,s'\sim\mathcal{P}} \left[ h_g(\xi,y) \right].$$

We also define constant $C_\delta := r_{\max} + (1+\gamma)\max\{R_{\max}, R_y\}$, and we immediately have

$$\|h_g(\xi,y)\|_2 \le |r(\xi) + \gamma\phi(s')^\top y - \phi(s)^\top y| \le r_{\max} + (1+\gamma)R_y \le C_g \tag{90}$$

and likewise, we have $\|\overline{h}_g(\xi,y)\|_2 \le C_g$.

The critic update can be written compactly as:

$$y_{k+1} = \Pi_{R_y} \left( y_k + \beta_k g(\xi_k, y_k) \right), \tag{91}$$

where $\xi_k := (s_k, a_k, s'_k)$ is the sample used to evaluate the stochastic gradient at $k$th update.

*Proof.* Using $y^*(\theta_k)$ as shorthand notation of $y^*_{\theta_k}$, we start with the optimality gap

$$\|y_{k+1} - y^*(\theta_{k+1})\|_2^2$$
$$= \|y_{k+1} - y^*(\theta_k) + y^*(\theta_k) - y^*(\theta_{k+1})\|_2^2$$
$$= \|y_{k+1} - y^*(\theta_k)\|_2^2 + \|y^*(\theta_k) - y^*(\theta_{k+1})\|_2^2 + 2\langle y_{k+1} - y^*(\theta_k), y^*(\theta_k) - y^*(\theta_{k+1})\rangle. \tag{92}$$

We first bound

$$\|y_{k+1} - y^*(\theta_k)\|_2^2 = \|\Pi_{R_y}\left(y_k + \beta_k g(\xi_k, y_k)\right) - y^*(\theta_k)\|_2^2$$
$$\le \|y_k + \beta_k g(\xi_k, y_k) - y^*(\theta_k)\|_2^2$$
$$= \|y_k - y^*(\theta_k)\|_2^2 + 2\beta_k \langle y_k - y^*(\theta_k), g(\xi_k, y_k)\rangle + \|\beta_k g(\xi_k, y_k)\|_2^2. \tag{93}$$

We first bound $\mathbb{E}[\langle y_k - y^*(\theta_k), g(\theta_k, y_k)\rangle | y_k]$ in (92) as

$$\mathbb{E}[\langle y_k - y^*(\theta_k), g(\theta_k, y_k)\rangle | y_k] = \left\langle y_k - y^*(\theta_k), \overline{h}_g(\theta_k, y_k) - \overline{h}_g(\theta_k, y^*(\theta_k)) \right\rangle$$
$$= \left\langle y_k - y^*(\theta_k), \mathbb{E}\left[ (\gamma\phi(s') - \phi(s))^\top (y_k - y^*(\theta_k))\phi(s) \right] \right\rangle$$
$$= \left\langle y_k - y^*(\theta_k), \mathbb{E}\left[ \phi(s) (\gamma\phi(s') - \phi(s))^\top \right] (y_k - y^*(\theta_k)) \right\rangle$$
$$= \left\langle y_k - y^*(\theta_k), A_{\pi_{\theta_k}} (y_k - y^*(\theta_k)) \right\rangle$$
$$\le -\lambda\|y_k - y^*(\theta_k)\|_2^2, \tag{94}$$

where the first equality is due to $\overline{h}_g(\theta, y^*_\theta) = A_{\theta,\phi}y^*(\theta) + b = 0$, and the last inequality follows Assumption 9.

Substituting (94) into (93), then taking expectation on both sides of (93) yields

$$\mathbb{E}\|y_{k+1} - y^*(\theta_k)\|_2^2 \le (1 - 2\lambda\beta_k)\mathbb{E}\|y_k - y^*(\theta_k)\|_2^2 + C_g^2\beta_k^2 \tag{95}$$

and plugging into into (92) yields

$$\mathbb{E}\|y_{k+1} - y^*(\theta_{k+1})\|_2^2 \le (1 - 2\lambda\beta_k)\mathbb{E}\|y_k - y^*(\theta_k)\|_2^2$$
$$+ 2\mathbb{E}\langle y_{k+1} - y^*(\theta_k), y^*(\theta_k) - y^*(\theta_{k+1})\rangle + \mathbb{E}\|y^*(\theta_k) - y^*(\theta_{k+1})\|_2^2 + C_g^2\beta_k^2. \tag{96}$$

Next we bound the third and fourth terms in (96) as

$$\mathbb{E}\langle y_{k+1} - y^*(\theta_k), y^*(\theta_k) - y^*(\theta_{k+1})\rangle$$
$$= \mathbb{E}\langle y_{k+1} - y^*(\theta_k), y^*(\theta_k) - y^*(\theta_{k+1}) - (\nabla y^*(\theta_k))^\top(\theta_{k+1} - \theta_k)\rangle$$
$$+ \mathbb{E}\langle y_{k+1} - y^*(\theta_k), (\nabla y^*(\theta_k))^\top(\theta_{k+1} - \theta_k)\rangle$$

$$\overset{(a)}{\le} \frac{L_{y,2}^2}{2}\mathbb{E}\|y_{k+1} - y^*(\theta_k)\|_2\|\theta_{k+1} - \theta_k\|_2^2 + \mathbb{E}\left[\langle y_{k+1} - y^*(\theta_k), \mathbb{E}[(\nabla y^*(\theta_k))^\top(\theta_{k+1} - \theta_k) \mid y_{k+1}]\rangle\right]$$

$$\overset{(b)}{\le} \frac{L_{y,2}^2}{2}\mathbb{E}\|y_{k+1} - y^*(\theta_k)\|_2\|\theta_{k+1} - \theta_k\|_2^2 + \alpha_k L_y\mathbb{E}\|y_{k+1} - y^*(\theta_k)\|\,\|\bar{h}_f(\theta_k, y_{k+1})\|$$

$$\overset{(c)}{\le} \frac{L_{y,2}^2}{4}\mathbb{E}\|y_{k+1} - y^*(\theta_k)\|_2^2\|\theta_{k+1} - \theta_k\|_2^2 + \frac{L_{y,2}^2}{4}\mathbb{E}\|\theta_{k+1} - \theta_k\|_2^2 + \alpha_k L_y\mathbb{E}\|y_{k+1} - y^*(\theta_k)\|\,\|\bar{h}_f(\theta_k, y_{k+1})\|$$

$$\overset{(d)}{\le} \frac{\alpha_k^2 C_f^2 L_{y,2}^2}{2}\mathbb{E}\|y_{k+1} - y^*(\theta_k)\|_2 + \frac{L_{y,2}^2}{4}\mathbb{E}\|\theta_{k+1} - \theta_k\|_2^2 + \alpha_k L_{y,2}^2\mathbb{E}\|y_{k+1} - y^*(\theta_k)\|_2^2 + \frac{\alpha_k}{4}\mathbb{E}\|\bar{h}_f(\theta_k, y_{k+1})\|^2$$

$$\le \left(\alpha_k L_{y,2}^2 + \frac{\alpha_k^2 C_f^2 L_{y,2}^2}{4}\right)\mathbb{E}\|y_{k+1} - y^*(\theta_k)\|_2^2 + \frac{\alpha_k}{4}\mathbb{E}\|\bar{h}_f(\theta_k, y_{k+1})\|^2 + \frac{\alpha_k^2 C_f^2 L_{y,2}^2}{4} \tag{97}$$

where (a) follows from the $L_{y,2}$-smoothness of $y^*$ with respect to $\theta$; (b) follows from $L_y$ is the Lipschitz constant of $y^*$ in Proposition 7 and

$$\mathbb{E}[(\nabla y^*(\theta_k))^\top(\theta_{k+1} - \theta_k) \mid y_{k+1}] = \nabla y^*(\theta_k))^\top\bar{h}_f(\theta_k, y_{k+1});$$

(c) uses the Young's inequality; (d) uses the Young's inequality and the fact that $\|\theta_{k+1} - \theta_k\|_2 = \alpha_k\|h_f(\xi_k', \theta_k, y_{k+1})\| \le C_g C_\psi = C_f$ and $\|\bar{h}_f(\theta_k, y_{k+1})\| \le C_f$.

We bound

$$\mathbb{E}\|y^*(\theta_k) - y^*(\theta_{k+1})\|_2^2 \le L_y^2\mathbb{E}\|\theta_k - \theta_{k+1}\|_2^2$$
$$\le L_y^2\alpha_k^2\mathbb{E}\left\|\hat{\delta}(\xi_k, y_k)\psi_{\theta_k}(s_k, a_k)\right\|_2^2 \le L_y^2 C_f^2\alpha_k^2 \tag{98}$$

where the inequality is due to the $L_y$-Lipschitz of $y^*(\theta)$ shown in Proposition 7, and the last inequality follows the fact that

$$\|\hat{\delta}(\xi_k, y_k)\psi_{\theta_k}(s_k, a_k)\|_2 \le C_g C_\psi = C_f. \tag{99}$$

Substituting (97)-(98) into (92) yields

$$\mathbb{E}\|y_{k+1} - y^*(\theta_{k+1})\|_2^2 \le \left(1 + \alpha_k L_{y,2}^2 + \frac{\alpha_k^2 C_f^2 L_{y,2}^2}{4}\right)\mathbb{E}\left[\|y_{k+1} - y^*(\theta_k)\|_2^2\right]$$
$$+ \frac{\alpha_k}{4}\mathbb{E}\|\bar{h}_f(\theta_k, y_{k+1})\|^2 + \frac{\alpha_k^2 C_f^2 L_{y,2}^2}{4} + L_y^2 C_f^2\alpha_k^2. \tag{100}$$

□

## D.3  Proof of Theorem 2

Recall the notations:

$$\hat{\delta}(\xi, y) := r(s, a, s') + \gamma\phi(s')^\top y - \phi(s)^\top y,$$
$$\bar{\delta}(\xi, y) := \mathbb{E}_{s\sim d_\theta, a\sim\pi_\theta, s'\sim\mathcal{P}}\left[r(s, a, s') + \gamma\phi(s')^\top y - \phi(s)^\top y \mid y\right]$$
$$\delta(\xi, \theta) := r(s, a, s') + \gamma V_{\pi_\theta}(s') - V_{\pi_\theta}(s).$$

The actor update can be written compactly as:

$$\theta_{k+1} = \theta_k + \alpha_k h_f(\xi_k', \theta_k, y_{k+1}) \tag{101}$$

where $h_f(\xi_k', \theta_k, y_{k+1}) := \hat{\delta}(\xi_k', y_{k+1})\psi_{\theta_k}(s_k, a_k)$. Define $\bar{h}_f(\theta_k, y_{k+1}) := \mathbb{E}[\hat{\delta}(\xi_k', y_{k+1})\psi_{\theta_k}(s_k, a_k)|y_{k+1}]$. Then we are ready to give the convergence proof.

*Proof.* From $L_F$-Lipschitz of policy gradient in Proposition 6, taking expectation conditioned on $\theta_k, y_{k+1}$, we have:

$$\mathbb{E}[F(\theta_{k+1})] - F(\theta_k) \tag{102}$$

$$\geq \mathbb{E}\langle\nabla F(\theta_k), \theta_{k+1} - \theta_k\rangle - \frac{L_F}{2}\mathbb{E}\|\theta_{k+1} - \theta_k\|_2^2$$

$$\geq \alpha_k\mathbb{E}\langle\nabla F(\theta_k), \bar{h}_f(\theta_k, y_{k+1})\rangle - \frac{L_F}{2}\mathbb{E}\|\theta_{k+1} - \theta_k\|_2^2$$

$$= \frac{\alpha_k}{2}\mathbb{E}\|\nabla F(\theta_k)\|^2 + \frac{\alpha_k}{2}\mathbb{E}\|\bar{h}_f(\theta_k, y_{k+1})\|^2 - \frac{\alpha_k}{2}\mathbb{E}\|\nabla F(\theta_k) - \bar{h}_f(\theta_k, y_{k+1})\|^2 - \frac{L_F}{2}\mathbb{E}\|\theta_{k+1} - \theta_k\|_2^2$$

$$\geq \frac{\alpha_k}{2}\mathbb{E}\|\nabla F(\theta_k)\|^2 + \frac{\alpha_k}{2}\mathbb{E}\|\bar{h}_f(\theta_k, y_{k+1})\|^2 - \frac{\alpha_k}{2}\mathbb{E}\|\nabla F(\theta_k) - \bar{h}_f(\theta_k, y_{k+1})\|^2$$

$$\qquad - \frac{L_F\alpha_k^2}{2}\mathbb{E}\|\bar{h}_f(\theta_k, y_{k+1})\|_2^2 - \frac{L_F\alpha_k^2}{2}\mathbb{E}\|\bar{h}_f(\theta_k, y_{k+1}) - h_f(\xi_k', \theta_k, y_{k+1})\|_2^2$$

$$\geq \frac{\alpha_k}{2}\mathbb{E}\|\nabla F(\theta_k)\|^2 + \left(\frac{\alpha_k}{2} - \frac{L_F\alpha_k^2}{2}\right)\mathbb{E}\|\bar{h}_f(\theta_k, y_{k+1})\|^2 - \frac{\alpha_k}{2}\mathbb{E}\|\nabla F(\theta_k) - \bar{h}_f(\theta_k, y_{k+1})\|^2 - \frac{L_F C_f^2\alpha_k^2}{2}$$

where the last inequality follows the definition of $C_f$ in (99).

We next bound the gradient bias as

$$\left\|\nabla F(\theta_k) - \bar{h}_f(\theta_k, y_{k+1})\right\|^2 = \left\|\nabla F(\theta_k) - \mathbb{E}[\hat{\delta}(\xi_k', y_{k+1})\psi_{\theta_k}(s_k, a_k)|y_{k+1}]\right\|^2$$

$$\leq 2\left\|\nabla F(\theta_k) - \mathbb{E}[\hat{\delta}(\xi_k', y^*(\theta_k))\psi_{\theta_k}(s_k, a_k)|y_{k+1}]\right\|^2$$

$$+ 2\left\|\mathbb{E}[(\hat{\delta}(\xi_k', y^*(\theta_k)) - \hat{\delta}(\xi_k', y_{k+1}))\psi_{\theta_k}(s_k, a_k)|y_{k+1}]\right\|^2$$

$$\leq 4\underbrace{\left\|\nabla F(\theta_k) - \mathbb{E}[\delta(\xi_k', \theta_k)\psi_{\theta_k}(s_k, a_k)|y_{k+1}]\right\|^2}_{I_1}$$

$$+ 4\underbrace{\left\|\mathbb{E}[\delta(\xi_k', \theta_k)\psi_{\theta_k}(s_k, a_k)|y_{k+1}] - \mathbb{E}[\hat{\delta}(\xi_k', y^*(\theta_k))\psi_{\theta_k}(s_k, a_k)|y_{k+1}]\right\|^2}_{I_2}$$

$$+ 2\underbrace{\left\|\mathbb{E}[(\hat{\delta}(\xi_k', y^*(\theta_k)) - \hat{\delta}(\xi_k', y_{k+1}))\psi_{\theta_k}(s_k, a_k)|y_{k+1}]\right\|^2}_{I_3}. \tag{103}$$

Then we bound $I_1$ as

$$I_1 = \|\nabla F(\theta_k) - \mathbb{E}[\delta(\xi_k', \theta_k)\psi_{\theta_k}(s_k, a_k)|\theta_k, y_{k+1}]\|^2$$

$$= \left\|\nabla F(\theta_k) - \mathbb{E}_{\substack{s_k \sim d_{\theta_k} \\ a_k \sim \pi_{\theta_k}, s_k' \sim \mathcal{P}}}\left[\left(r(s_k, a_k, s_k') + \gamma V_{\pi_{\theta_k}}(s_k') - V_{\pi_{\theta_k}}(s_k)\right)\psi_{\theta_k}(s_k, a_k)\Big|\theta_k, y_{k+1}\right]\right\|^2$$

$$= \left\|\nabla F(\theta_k) - \mathbb{E}_{\substack{s_k \sim d_{\theta_k} \\ a_k \sim \pi_{\theta_k}}}\left[A_{\pi_{\theta_k}}(s_k, a_k)\psi_{\theta_k}(s_k, a_k)\Big|\theta_k, y_{k+1}\right]\right\|^2 = 0$$

where the last equality follows from the policy gradient theorem.

Then we bound $I_2$ as

$$
\begin{aligned}
I_2 &= \left\| \mathbb{E}[\delta(\xi_k', \theta_k)\psi_{\theta_k}(s_k, a_k)|\theta_k, y_{k+1}] - \mathbb{E}[\hat{\delta}(\xi_k', y^*(\theta_k))\psi_{\theta_k}(s_k, a_k)|\theta_k, y_{k+1}] \right\|^2 \\
&= \left\| \mathbb{E}\left[ \left(\delta(\xi_k', \theta_k) - \hat{\delta}(\xi_k', y^*(\theta_k))\right)\psi_{\theta_k}(s_k, a_k)|\theta_k, y_{k+1}\right] \right\|^2 \\
&= \left\| \mathbb{E}\left[ \left|\delta(\xi_k', \theta_k) - \hat{\delta}(\xi_k', y^*(\theta_k))\right| \|\psi_{\theta_k}(s_k, a_k)\| |\theta_k, y_{k+1}\right] \right\|^2 \\
&= C_\psi^2 \left\| \mathbb{E}\left[ \left|\delta(\xi_k', \theta_k) - \hat{\delta}(\xi_k', y^*(\theta_k))\right| |\theta_k, y_{k+1}\right] \right\|^2 \\
&\leq C_\psi^2 \left( \gamma \mathbb{E}\left|\phi(s_k')^\top y^*(\theta_k) - V_{\pi_{\theta_k}}(s_k')\right| + \mathbb{E}\left|V_{\pi_{\theta_k}}(s_k) - \phi(s_k)^\top y^*(\theta_k)\right| \right) \\
&\leq C_\psi^2 \left( \gamma \sqrt{\mathbb{E}\left|\phi(s_k')^\top y^*(\theta_k) - V_{\pi_{\theta_k}}(s_k')\right|^2} + \sqrt{\mathbb{E}\left|V_{\pi_{\theta_k}}(s_k) - \phi(s_k)^\top y^*(\theta_k)\right|^2} \right) \\
&\leq C_\psi^2 (1 + \gamma)\epsilon_{app}.
\end{aligned}
$$

Then we bound $I_3$ as

$$
\begin{aligned}
I_3 &= \left\| \mathbb{E}[(\hat{\delta}(\xi_k', y^*(\theta_k)) - \hat{\delta}(\xi_k', y_{k+1}))\psi_{\theta_k}(s_k, a_k)|\theta_k, y_{k+1}] \right\|^2 \\
&\leq C_\psi^2 \mathbb{E}\left[ \|\hat{\delta}(\xi_k', y^*(\theta_k)) - \hat{\delta}(\xi_k', y_{k+1})\|^2 |\theta_k, y_{k+1}\right] \\
&= C_\psi^2 \mathbb{E}\left[ \|\gamma\phi(s_k')^\top y^*(\theta_k) - \phi(s_k)^\top y^*(\theta_k) - \gamma\phi(s_k')^\top y_{k+1} + \phi(s_k)^\top y_{k+1}\|^2 |\theta_k, y_{k+1}\right] \\
&\leq C_\psi^2 (1 + \gamma)\|y^*(\theta_k) - y_{k+1}\|^2.
\end{aligned}
$$

Then (103) can be rewritten as

$$
\left\| \nabla F(\theta_k) - \bar{h}_f(\theta_k, y_{k+1}) \right\|^2 \leq 4C_\psi^2 (1 + \gamma)\epsilon_{app} + 2C_\psi^2 (1 + \gamma)\|y^*(\theta_k) - y_{k+1}\|^2
$$

plugging which into (102) leads to

$$
\begin{aligned}
\mathbb{E}[F(\theta_{k+1})] &\geq F(\theta_k) + \frac{\alpha_k}{2}\mathbb{E}\|\nabla F(\theta_k)\|^2 + \left(\frac{\alpha_k}{2} - \frac{L_F \alpha_k^2}{2}\right)\mathbb{E}\left\|\bar{h}_f(\theta_k, y_{k+1})\right\|^2 \\
&\quad - \frac{\alpha_k}{2}\mathbb{E}\left\|\nabla F(\theta_k) - \bar{h}_f(\theta_k, y_{k+1})\right\|^2 - \frac{L_F C_f^2 \alpha_k^2}{2} \\
&\geq F(\theta_k) + \frac{\alpha_k}{2}\mathbb{E}\|\nabla F(\theta_k)\|^2 + \left(\frac{\alpha_k}{2} - \frac{L_F \alpha_k^2}{2}\right)\mathbb{E}\left\|\bar{h}_f(\theta_k, y_{k+1})\right\|^2 - \frac{L_F C_f^2 \alpha_k^2}{2} \\
&\quad - 2\alpha_k C_\psi^2 (1 + \gamma)\epsilon_{app} - \alpha_k C_\psi^2 (1 + \gamma)\|y^*(\theta_k) - y_{k+1}\|^2. \tag{104}
\end{aligned}
$$

Consider the difference of the Lyapunov function $\mathbb{V}^k := -F(\theta_k) + \|y_k - y^*(\theta_k)\|_2^2$, given by

$$
\begin{aligned}
\mathbb{E}[\mathbb{V}^{k+1}] - \mathbb{E}[\mathbb{V}^k] &= -\mathbb{E}[F(\theta_{k+1})] + \mathbb{E}\|y_{k+1} - y^*(\theta_{k+1})\|_2^2 + \mathbb{E}[F(\theta_k)] - \mathbb{E}\|y_k - y^*(\theta_k)\|_2^2 \\
&\leq -\frac{\alpha_k}{2}\mathbb{E}\|\nabla F(\theta_k)\|^2 - \left(\frac{\alpha_k}{2} - \frac{L_F \alpha_k^2}{2}\right)\mathbb{E}\left\|\bar{h}_f(\theta_k, y_{k+1})\right\|^2 + \frac{L_F C_f^2 \alpha_k^2}{2} + 2\alpha_k C_\psi^2 (1 + \gamma)\epsilon_{app} \\
&\quad + \alpha_k C_\psi^2 (1 + \gamma)\|y^*(\theta_k) - y_{k+1}\|^2 + \mathbb{E}\|y_{k+1} - y^*(\theta_{k+1})\|_2^2 - \mathbb{E}\|y_k - y^*(\theta_k)\|_2^2 \\
&\leq -\frac{\alpha_k}{2}\mathbb{E}\|\nabla F(\theta_k)\|^2 - \left(\frac{\alpha_k}{2} - \frac{\alpha_k}{4} - \frac{L_F \alpha_k^2}{2}\right)\mathbb{E}\left\|\bar{h}_f(\theta_k, y_{k+1})\right\|^2 + \frac{L_F C_f^2 \alpha_k^2}{2} \\
&\quad + \left(1 + \alpha_k L_{y,2}^2 + \frac{\alpha_k^2 C_f^2 L_{y,2}^2}{4} + \alpha_k C_\psi^2 (1 + \gamma)\right)\mathbb{E}\left[\|y_{k+1} - y^*(\theta_k)\|_2^2\right] \\
&\quad - \mathbb{E}\|y_k - y^*(\theta_k)\|_2^2 + \frac{\alpha_k^2 C_f^2 L_{y,2}^2}{4} + L_y^2 C_f^2 \alpha_k^2 + 2\alpha_k C_\psi^2 (1 + \gamma)\epsilon_{app}. \tag{105}
\end{aligned}
$$

Applying (95) to bound $\mathbb{E}\left[\|y_{k+1} - y^*(\theta_k)\|_2^2\right]$, we have

$$\mathbb{E}[\mathbb{V}^{k+1}] - \mathbb{E}[\mathbb{V}^k]$$

$$\leq -\frac{\alpha_k}{2}\mathbb{E}\|\nabla F(\theta_k)\|^2 - \left(\frac{\alpha_k}{2} - \frac{\alpha_k}{4} - \frac{L_F\alpha_k^2}{2}\right)\mathbb{E}\left\|\bar{h}_f(\theta_k, y_{k+1})\right\|^2 + \frac{L_F C_f^2 \alpha_k^2}{2} + 2\alpha_k C_\psi^2(1+\gamma)\epsilon_{app}$$

$$+ \left[\left(1 + \alpha_k L_{y,2}^2 + \frac{\alpha_k^2 C_f^2 L_{y,2}^2}{4} + \alpha_k C_\psi^2(1+\gamma)\right)(1 - 2\lambda\beta_k) - 1\right]\mathbb{E}\|y_k - y^*(\theta_k)\|_2^2$$

$$+ \left(1 + \alpha_k L_{y,2}^2 + \frac{\alpha_k^2 C_f^2 L_{y,2}^2}{4} + \alpha_k C_\psi^2(1+\gamma)\right)C_g^2\beta_k^2 + \frac{\alpha_k^2 C_f^2 L_{y,2}^2}{4} + L_y^2 C_f^2 \alpha_k^2. \tag{106}$$

Similar to the steps (54)-(56), if we select

$$\alpha_k = \min\left\{\frac{1}{2L_F}, \frac{\alpha}{\sqrt{K}}\right\}, \qquad \beta_k = \frac{4L_{y,2}^2 + 8C_\psi^2 + C_f^2 L_{y,2}^2/2L_F}{8\lambda}\alpha_k. \tag{107}$$

which ensures that

$$\frac{\alpha_k}{4} - \frac{L_F\alpha_k^2}{2} \geq 0 \tag{108a}$$

$$\left(1 + \alpha_k L_{y,2}^2 + \alpha_k C_\psi^2(1+\gamma) + \frac{\alpha_k^2 C_f^2 L_{y,2}^2}{4}\right)(1 - 2\lambda\beta_k) \leq 1 \tag{108b}$$

we can simplify (106) as

$$\mathbb{E}[\mathbb{V}^{k+1}] - \mathbb{E}[\mathbb{V}^k] \leq -\frac{\alpha_k}{2}\mathbb{E}\|\nabla F(\theta_k)\|^2 + \frac{L_F C_f^2 \alpha_k^2}{2} + 2\alpha_k C_\psi^2(1+\gamma)\epsilon_{app} + \frac{\alpha_k^2 C_f^2 L_{y,2}^2}{4}$$

$$+ \left(1 + \alpha_k L_{y,2}^2 + \frac{\alpha_k^2 C_f^2 L_{y,2}^2}{4} + \alpha_k C_\psi^2(1+\gamma)\right)C_g^2\beta_k^2 + L_y^2 C_f^2 \alpha_k^2. \tag{109}$$

After telescoping, we have

$$\frac{1}{K}\sum_{k=1}^{K}\mathbb{E}\|\nabla F(\theta_k)\|^2 \leq \frac{2\mathbb{V}^1}{\alpha_k K} + L_F C_f^2 \alpha_k + 4C_\psi^2(1+\gamma)\epsilon_{app} + \frac{\alpha_k C_f^2 L_{y,2}^2}{2} + 2L_y^2 C_f^2 \alpha_k$$

$$+ 2\left(1 + \alpha_k L_{y,2}^2 + \frac{\alpha_k^2 C_f^2 L_{y,2}^2}{4} + \alpha_k C_\psi^2(1+\gamma)\right)\frac{C_g^2\beta_k^2}{\alpha_k} \tag{110}$$

which, together with $\alpha_k = \mathcal{O}(1/\sqrt{K}), \beta_k = \mathcal{O}(1/\sqrt{K})$, completes the proof. $\qquad\square$