# OpenReview forum: "Closing the Gap: Tighter Analysis of Alternating Stochastic Gradient Methods for Bilevel Problems"
_NeurIPS.cc/2021/Conference — NeurIPS 2021 Spotlight_

### Official Review · Reviewer_WgWk · 2021-07-05

**Rating:** 8
**Confidence:** 4

**Summary:**

The paper proposes a tighter analysis of the complexity of alternating gradient descent on stochastic bilevel optimization problems. The tighter analysis is then applied on several nested optimization problems like min-max optimization, compositional problems as well as actor critic methods for MDP problems. A key step in the analysis is leveraging the smoothness of the lower optimization problem with respect to the decision variables of the upper level problem.

**Limitations And Societal Impact:**

Limitations and negative social impact are adequately discussed.

**Main Review:**

Overall I think the paper is well written and the idea of proposing a unified analysis for all the optimization problems discussed in this work can have significant impact. Also closing the gap between the complexity of simple SGD and bilevel optimization is definitely important in the process of delineating which aspects of bilevel optimization are actually hard.

While the smoothness of the lower level problem has been used in prior work (e.g. [27]), it is the smoothness of the gradients proved in Lemma 2 that opens the way to the improved rate. While Lemma 2 itself is rather straightforward to prove,  how and why it can be used in the proof is definitely not. Small tweaks to existing approaches are not sufficient for the reasons summarized under Equation 20. I suggest to the authors to expand more on this reasons in the Appendix to make the paper more self contained.

Overall a very strong and well written submission.

**Time Spent Reviewing:**

5

---

> ### Author Response · Authors · 2021-08-10
> **Response to Reviewer WgWk**
>
> We thank the reviewer for acknowledging that our idea of proposing a unified analysis for all the optimization problems discussed in this work can have significant impact. Below is our reponse to your minor comment.
>
> **Q1. While Lemma 2 itself is rather straightforward to prove, how and why it can be used in the proof is definitely not. I suggest to the authors to expand more on this reasons in the Appendix to make the paper more self contained.**
>
> As the reviewer correctly pointed out, the smoothness of the gradients proved in Lemma 2 opens the way to the improved rate. Following your suggestion, we will further discuss how to use this smoothness in the proof and why it can lead to the improved rates. We believe this technique can be used in the future analysis of stochastic nested problems.

---

### Official Review · Reviewer_PMym · 2021-07-13

**Rating:** 8
**Confidence:** 3

**Summary:**

This paper studies the problem of solving a stochastic bilevel optimization problem, where the upper level optimization problem depends on the solution of the lower level optimization problem. The authors show that this stochastic bilevel optimization problem covers three different classes of problems (stochastic compositional, min-max, and bilevel optimization). The authors then unify the three SGD-type methods for the respective problems into a single approach called ALSET. Besides, compared with previous work, they also improve the sample complexity from $\mathcal{O}(\epsilon^{-2.5})$ to $\mathcal{O}(\epsilon^{-2})$.

**Limitations And Societal Impact:**

There is no negative societal impact.

**Main Review:**

I found the manuscript to be clearly written and technically sound. The proposed algorithms are also easy to understand. The authors give a detailed comparison between their result and previous results, where the improvement in the sample complexity is clearly stated. The only suggestion I have is can you also provide some numerical experiments to support your theoretical results?

**Time Spent Reviewing:**

2 hrs

---

> ### Author Response · Authors · 2021-08-10
> **Response to Reviewer PMym**
>
> Thank you very much for acknowledging that our paper is clearly written and technically sound. We would like to address your only comment below.
>
> **Q1. The only suggestion I have is can you also provide some numerical experiments to support your theoretical results?**
>
> Following your valuable suggestion, during the rebuttal period, we have conducted the experiment using the risk-averse portfolio management task on a benchmark dataset - 100 Book-to-Market. This is a typical application of stochastic nested optimization used in references [39], [40]. We compared the two-timescale nested SGD approach (SCGD) with our single-timescale nested SGD approach ALSET.
>
> We tune the stepsizes $\alpha, \beta$ by following the suggested order in the original SCGD paper and then using a grid search for the constant $c$, that is
>
> SCGD: $\alpha = c\cdot k^{-3/4}$, $\beta = c\cdot k^{-1/2}$
>
> ALSET: $\alpha = c\cdot  k^{-1/2}$, $\beta = c\cdot k^{-1/2}$
>
> The constant $c$ is chosen from the searching grid  \{$10^{-3}, 5\times 10^{-4}, 10^{-4}$\} and optimized for each algorithm in terms of ergodic average gradient norm versus the number of iterations. In the table below, we report the logarithmic value of the average gradient norm performance of SCGD and ALSET with both the above decreasing stepsizes and the constant stepsizes (replacing k with total number of iterations K=1000). We can observe that the empirical gradient-norm convergence rate of ALSET is no worse than the worst-case theoretical rate, and ALSET outperforms SCGD thanks to its single-timescale stepsizes.
>
>
> | \# of iter | log(iter) |SCGD| ALSET | ALSET (constant)|
> | :------------- | :------------- | :----------: | :-----------: | :-----------: |
> |  10  | 2.30|  5.32   |  5.31    |  5.63 |
> | 100 | 4.61 |  3.78| 3.49   | 3.63 |
> | 200| 5.30   |  3.40   | 2.94  |  3.06 |
> | 1000| 6.91   | 2.57  |  1.65 |  2.06|
>
> Thanks again for summarizing our paper and recognizing the merits of our paper!

---

> > ### Comment · Reviewer_PMym · 2021-08-20
> > **Response to authors**
> >
> > Thank you for providing the numerical experiments.

---

### Official Review · Reviewer_vxWe · 2021-07-15

**Rating:** 8
**Confidence:** 3

**Summary:**

This paper studies the altenating gradient descent method for stochastic nested problems.  It uses a general framework which contains the compositional gradient descent, the minimax problem, and the actor-critic method. It also provides a tighter analysis of the convergence result and better convergence rates in many cases.

**Limitations And Societal Impact:**

The authors addressed the limitations adequately.

**Main Review:**

The contributions of the work is mainly theoretical.  Although the proof technique is fairly standard, it achieves better rates with a simple single-simescale algorithm. This work also points out the importance of the hidden smoothness of the optimizer of the lower level. It can be very important for further works in the field. The general framework allows us to easily generalize different variants of gradient descent methods to different settings. As a result, there are many possible lines of research following this work.

The paper is clearly written and the results are well-organized. Previous works are cited properly.

**Time Spent Reviewing:**

4

---

> ### Author Response · Authors · 2021-08-10
> **Response to Reviewer vxWe**
>
> Thanks a lot for summarizing the contributions of our paper. They are very accurate.
>
> Also thanks for recognizing the merits and the potential impact of our theoretical results!
>
> We very much appreciate your time and efforts on reviewing our paper.

---

### Official Review · Reviewer_TdaR · 2021-07-16

**Rating:** 7
**Confidence:** 4

**Summary:**

This paper considers the problem of stochastic nested optimization which unified the stochastic compositional optimization and stochastic min-max optimization. It presents a tighter theoretical analysis for classical algorithm which uses vanilla SGD-type update. The sample complexity of this improved analysis matches those with modifications.

**Limitations And Societal Impact:**

None.

**Main Review:**

This paper is well-written. I have some comments below.

1. This paper improves the sample complexity for SGD-type algorithms to solve stochastic nested optimization. The algorithm and some results (like observation (6) and estimation (8)) are already presented in previous literature (e.g., Ghadimi and Wang (2018)). Theorem 1 gives a tighter bound of such algorithm where they utilize the observations in Lemma 2. The authors give a clear presentation of their technique and highlight the key steps of the proof towards Theorem 1. However, it is my understanding that the novelty of this paper is merely the observation of lemma 2 and the tighter analysis following it.

2. Under the unified structure, they extend their theoretical results to stochastic compositional optimization as well as stochastic min-max optimization. They also give an application example of the actor-critic method with linear value approximation.

3. However, this paper is not complete. A conclusion part should be presented.

References mentioned above:

1. Ghadimi, Saeed, and Mengdi Wang. "Approximation methods for bilevel programming." *arXiv preprint arXiv:1802.02246* (2018).

**Time Spent Reviewing:**

3

---

> ### Author Response · Authors · 2021-08-10
> **Response to Reviewer TdaR**
>
> Thanks a lot for the careful review and favorable recommendation.
>
> **Q1. The novelty of this paper is merely the observation of lemma 2 and the tighter analysis following it.**
>
> Indeed, it is the smoothness in Lemma 2 that opens the way to the improved rate. However, as one of  reviewers has pointed out, how and why it can be used in the proof is definitely not strightforward. We will expand more on this technique and hope it can be used in future proofs.
>
> **Q2. However, this paper is not complete. A conclusion part should be presented.**
>
> Due to space limitation, we cannot add the conclusion in the initial submission. But following your suggestion, we will certainly add a conclusion part in the final paper. We provide a tentative conclusion below.
>
> > ``This paper unifies several SGD-type approaches for stochastic nested problems into a single nested SGD approach that we term ALSET, which runs in a single timescale and uses a fixed batch size. The paper further presents a tighter analysis for using it to solve three different stochastic nested problems - stochastic compositional, min-max and bilevel problems.
> Under the new analysis, to achieve an $\epsilon$-stationary point of the nested problem, ALSET requires ${\cal O}(\epsilon^{-2})$ samples in total, which is the best sample complexity of the vanilla nested SGD. Under certain regularity conditions, applying the new analysis to an alternating version of the actor-critic algorithm also yields the state-of-the-art sample complexity.
>
> > To overcome the limitations of ALSET, our future work consists of performing comprehensive simulations to validate this sample complexity in practice and relaxing the regularity conditions needed to achieve this theoretical result.’’
>
>
> Any comments and suggestions on the conclusion are welcome.

---

### Decision · Program_Chairs · 2021-09-28

**Decision:**

Accept (Spotlight)

**Comment:**

It is the consensus of the reviewers that this paper makes a worthwhile contribution to stochastic optimization. The analysis of alternating gradient descent is tight. The implications of the analysis on stochastic compositional, min-max, and bilevel optimizations are interesting and useful.

**Consistency Experiment:**

NeurIPS has a long history of experimentation. In 2014, NeurIPS ran an experiment in which 10% of submissions were reviewed by two independent committees to quantify the randomness in the review process. This year, we repeated a variant of this experiment to see how the quality of the review process has changed over time.  This paper was part of the experiment and was therefore assigned to two committees (consisting of reviewers, an Area Chair, and a Senior Area Chair) that reached independent decisions.  If both committees made the same recommendation, this recommendation was followed. If a single committee recommended acceptance, the paper was accepted (with the exception of a few cases in which the other committee identified what we considered a fatal flaw, e.g., an error in a key result).

This copy’s committee reached the following decision: **Accept (Spotlight)**

The other committee assigned to the paper recommended **Accept (Poster)**.  You can find the other set of reviews, along with any follow up discussion with the authors here:
https://openreview.net/forum?id=r6cNUjS8cm0